# Environmental DNA provides higher resolution assessment of riverine biodiversity and ecosystem function via spatio-temporal nestedness and turnover partitioning

Mathew Seymour [1✉], François K. Edwards [2], Bernard J. Cosby[3], Iliana Bista[4,5], Peter M. Scarlett[2], Francesca L. Brailsford [6], Helen C. Glanville[6], Mark de Bruyn [7], Gary R. Carvalho[6] & Simon Creer [6]

Rapidly assessing biodiversity is essential for environmental monitoring; however, traditional approaches are limited in the scope needed for most ecological systems. Environmental DNA (eDNA) based assessment offers enhanced scope for assessing biodiversity, while also increasing sampling efficiency and reducing processing time, compared to traditional methods. Here we investigated the effects of landuse and seasonality on headwater community richness and functional diversity, via spatio-temporal dynamics, using both eDNA and traditional sampling. We found that eDNA provided greater resolution in assessing biodiversity dynamics in time and space, compared to traditional sampling. Community richness was seasonally linked, peaking in spring and summer, with temporal turnover having a greater effect on community composition compared to localized nestedness. Overall, our assessment of ecosystem function shows that community formation is driven by regional resource availability, implying regional management requirements should be considered. Our findings show that eDNA based ecological assessment is a powerful, rapid and effective assessment strategy that enables complex spatio-temporal studies of community diversity and ecosystem function, previously infeasible using traditional methods.

[1] Department of Ecology, Swedish University of Agricultural Sciences, Uppsala, Sweden. [2] Centre for Ecology & Hydrology, Wallingford, UK. [3] Centre for Ecology & Hydrology, Environment Centre Wales, Bangor, UK. [4] Department of Genetics, University of Cambridge, Cambridge, UK. [5] Wellcome Sanger Institute, Hinxton, UK. [6] School of Natural Sciences, Bangor University, Bangor, UK. [7] The University of Sydney, School of Life and Environmental Sciences, Sydney, Australia. ✉email: mathew.seymour@slu.se

Modern human development has drastically increased the speed at which we alter our physical and societal environments, which have rapid and drastic effects on our ecosystems and their functions. We are often too slow to act on key changes in our environment, which makes the recovery and rehabilitation of healthy ecosystems more costly compared to programs that actively monitor ecosystems[1]. Active ecosystem monitoring relies greatly on monitoring the change in biological communities (e.g. biodiversity) to assess ecosystem function and health[2,3]. Despite the growing call to safeguard our natural ecosystems we are currently experiencing a major decline in global biodiversity, further highlighting the inability of current capabilities to actively monitor and respond to these threats[4]. It is therefore paramount that we develop more effective biodiversity assessment practices to increase our understanding of complex ecological systems and to promote ecosystem function and health[5].

Accurate assessment of biodiversity relies on our understanding of the localized, spatial and temporal processes that shape changes in biodiversity in time and space[6–8]. Localized (i.e. site-specific) biodiversity assessments dominate current monitoring practices, whereby community composition is used to infer local environmental conditions. As changes in biodiversity are also influenced by spatial (e.g. dispersal) and temporal (e.g. phenology) factors; however, it is also essential to assess temporal and spatial community dynamics[6]. Current assessment practices can also be improved by assessing the functional response of communities to changes in environmental condition, which would provide a clear causal link to what aspects of the environment are altering community composition[2].

Biodiversity can be quantified in many different ways, including through assessment of community richness or functional diversity. Richness is the most common metric of biodiversity and is defined as the number of unique taxonomic units per site/sample. Richness is often positively correlated with environmental heterogeneity, which is often attributed to greater levels of functional diversity[9–11]. Alternatively, functional diversity directly quantifies the functionally disparate taxa within a community, and is becoming increasingly recognized as an important component of effective biomonitoring[12]. Additionally, differences in localized (e.g. sampling locations) biodiversity measures (e.g. richness) between communities; in space or time, commonly referred to as beta-diversity, is used to assess whether changes in biodiversity are influenced by more local or spatial factors[13]. Partitioning the variance of beta-diversity into nested and turnover components provides even greater insight into the processes that are driving inter-community homogenization or differentiation[14]. In contrast to existing assessment practices, additional sampling of within and among sites is required to effectively incorporate rapid spatio-temporal biodiversity assessment. Unfortunately, traditional monitoring methodologies are often limited or forced to simplify methods to cope with limited computational power or to reduce cost[15]. To implement increased spatial and temporal biodiversity assessment we have to develop and utilize improved biodiversity assessment methodologies, to generate the data needed to rapidly assess ecosystems, particularly at increased spatial and temporal resolution.

The application of environmental DNA (eDNA) and metabarcoding has been shown to increase the sampling resolution for biodiversity assessment efforts[16–18], though for some species-specific studies, specialized traditional methods may outperform eDNA surveys[19,20]. Environmental DNA is extracted directly from an environmental sample (e.g. water, soil, or air) without prior isolation of the organisms themselves[21,22]. Sources of eDNA include sloughed skin cells, urine, feces, saliva, or other bodily secretions, and consist of both free molecules (extracellular DNA)

and cells[23–25]. Furthermore, eDNA collected from water samples has highly sensitive detection capability, and provides a wider sampling application (e.g. substrate or surface area conditions) with lower environmental impact, compared to traditional methods. There is a wide range of reported efficiencies between eDNA studies and traditional sampling methods, however, as the number of eDNA studies increases, so does our ability to account for random variability in eDNA biodiversity data[18]. Combined with high throughput sequencing (HTS) applications, eDNA sampling is rapidly being integrated into standard ecological monitoring practices, including assessments of population and communities across spatial and temporal scales for rivers[16,26,27], lakes[28–30], and marine environments[31–33]. What is currently lacking is the link between functional and community diversity dynamics using eDNA across appropriate spatial and temporal scales. To develop and validate an eDNA-based approach to biodiversity assessment it is important to develop hypotheses from our current understanding of functional and community principles and dynamics.

Headwater riverine biodiversity is one of the longest standing realms of ecology and a key component of current freshwater biomonitoring and assessment[34]. Riverine catchments are a crucial component of regional biodiversity that harbor high levels of diversity due to their hierarchical structure, environmentally diverse habitats, and unique headwater communities[34–36]. Current biomonitoring practices in rivers utilize biological indices derived from freshwater macroinvertebrates[37], as their localized community assembly dynamics are strongly linked to environmental conditions[34,38]. Freshwater macroinvertebrates represent a wide range of species and functional groups, which respond dynamically to temporal and spatial environmental filtering, thus providing a clear depiction of localized ecosystem function[36,39,40]. Specifically, functional feeding groups of freshwater macroinvertebrates allow for direct assessments of nutrient cycling, productivity, and decomposition[40]. Traditional taxonomic identification of freshwater macroinvertebrates, however, is largely limited to mature life stages that can be difficult to identify or differentiate among similar species or genera[37]. The high level of taxonomic specialization required to identify specimens and the long processing times per sample renders large-scale ecosystem-wide traditional assessments expensive and time consuming[15,41]. In response, there is currently an ongoing rapid push to implement eDNA based riverine biodiversity assessment practices[16,22,42], which forms the basis for this study.

Here, we assess seasonal patterns of biodiversity and functional diversity, using an experimental design that utilizes headwater sampling sites to associate local environmental conditions within the same environmentally heterogeneous geographic region (i.e. catchment; Fig. 1). We utilized a combined eDNA and traditional based biodiversity assessment approach to allow for direct comparison between traditional ecological expectations and molecular based, eDNA methods. We investigated four main objectives and hypotheses. One, riverine macroinvertebrate biodiversity, specifically localized community richness, is expected to peak during spring and summer, when many stream macroinvertebrates are emerging as adults and reproducing, compared to fall and winter months when total biomass of many species has declined[28,43]. Two, eDNA biodiversity will be greater compared to traditional sampling, following previous experimental findings[16]. Three, utilizing the nested and turnover components of inter-community similarity (i.e. beta-diversity), we can expect high turnover within sites as community assembly changes over time, and high nestedness across sampling sites, attributed to environmental filtering. Alternatively, low nestedness could indicate a low effect of environmental filtering and a greater effect of stochastic or biotic factors influencing the localized community

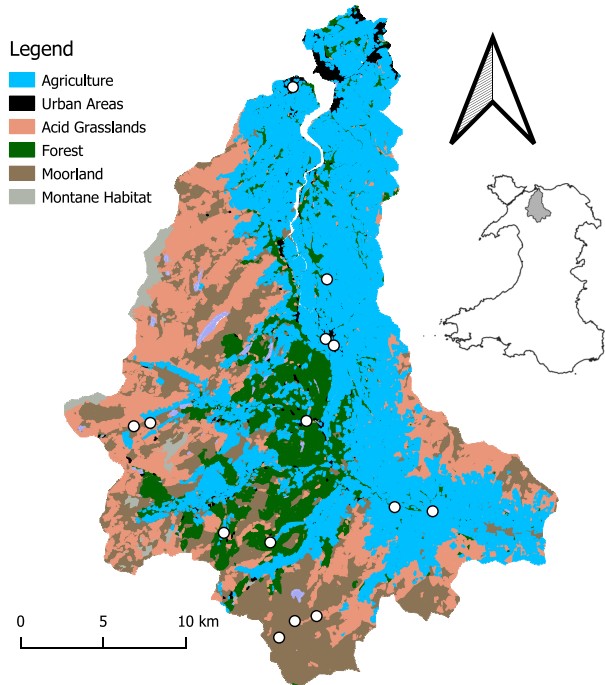

**Fig. 1 Map of the study area (Conwy Catchment, North Wales).** Colors colors correspond to different landuse types, including agriculture (blue), bog/moorland (brown), acid grassland (pink), forested (green), and urban areas (black). White circles indicate sampling locations ($N = 14$).

**Table 1 Environmental site variables (first column) with the associated mean, standard deviation, min and max across sampling sites.**

|  | Units | Mean | SD | Min | Max |
|---|---|---|---|---|---|
| pH |  | 6.87 | 0.67 | 5.17 | 7.55 |
| Conductivity | µS/cm | 92.21 | 69.98 | 25.50 | 252.50 |
| Depth | cm | 17.87 | 8.84 | 6.33 | 37.00 |
| Moss | % | 8.61 | 9.46 | 0.25 | 31.25 |
| Algae | % | 2.32 | 4.31 | 0.00 | 15.00 |
| Plant | % | 1.16 | 1.26 | 0.00 | 3.75 |
| Boulder | % | 51.02 | 15.31 | 18.33 | 73.00 |
| Gravel | % | 42.69 | 12.12 | 26.25 | 65.75 |
| Sand | % | 6.06 | 4.28 | 0.75 | 18.33 |

assemblies. Four, environmental filtering effects that are linked to habitat modification, such as agriculture or urbanized areas, are expected to negatively impact macroinvertebrate diversity and functionality, particularly Chironomidae, and Ephemeroptera, Plecoptera, and Trichoptera (EPT) taxa, indicating variable site-specific ecosystem health across the region[44].

We found eDNA biodiversity to be a better descriptor of the total macroinvertebrate diversity across all sampling sites, with trends between the two methods showing general similarities. Community richness was greater in the spring and summer and lowest during the winter, as expected, with the greatest change in community composition across seasons linked to changes in Chironomidae genera richness. Landuse, while showing distinct environmental differentiation, was not associated with local community richness. Additionally, spatio-temporal dynamics among communities were found to be predominately turnover driven, indicating strong seasonal or region-wide effects. Nestedness effects were mostly limited, suggesting weak localized environmental sorting of the communities. Lastly, functional diversity showed clear region-wide generalization of feeding functionality, suggesting biodiversity is driven by regional-based bottom-up dynamics, which suggest biodiversity management should focus on regional over localized spatial extents.

## Results

**Sequencing results.** After stringent filtering and quality control, 12,592,362 reads were obtained with an average of 74,954 (±31,050) reads per sample. Negative controls, which showed no bands on agarose gels post library preparation, generated 676 reads across all blanks ($N = 12$). Of the negative control reads, 411 reads were unknown bacteria, 3 reads were associated with three genera of Rhodophyta (red algae), 2 reads were linked to unknown fungi, and 260 reads were linked to a single Dipteran ASV across four blanks. For downstream analyses, the Dipteran ASV was removed from subsequent analyses, and all other

potential contaminants not included, as they were non-targeted. In total, 20,437 ASVs were identified. Average reads per site, after rarefaction was 75,871 (±37,670) with four sites having less than 10,000 reads from four different sites across three different landuse types and two seasons. The average number of taxonomic assignments per site was 13,260 ASVs (±9567). The average number of kick-net sampling specimens per site was 1529 individuals (±1555). Mean singletons per site was 9.86 (±6.67) for eDNA and 9.02 (±3.29) for kick-net sampling. Singletons were included in subsequent analyses given the robust use of sample replication used in the study design, whereby if a sequence was not observed in at least 2 of the 3 replicate samples the sequence was not included in the downstream analysis.

**Community dynamics.** All environmental variables and their associated summary statistics are presented in Table 1 and Fig. 2. Overall, we observed 226 unique genera using the eDNA based approach and 83 genera using the traditional kick-netting approach (Table 2). On average, eDNA genera accounted for 78.2% of the unique observed diversity in a given site, with traditional methods accounting for 5.9%, with an overlap between the two methods of 15.9% (Fig. 3). Key differences between the methods were the higher number of genera observed using eDNA vs traditional methods for Chironomidae (75 vs 10), Oligochaeta (23 vs 2), Trichoptera (33 vs 24), Rotifera (8 vs 0), Coleoptera (20 vs 14) and Copepoda (5 vs 0) (Table 2). The full breakdown of genera per landuse type per sampling method can be found in Supplementary Data 1.

Genera, Chironomidae, EPT and functional richness, derived from eDNA, were all significantly greater than traditionally sampled richness ($p < 0.001$) (Table 3 and Fig. 3). Genera richness differed significantly across season ($p = 0.002$) and landuse type ($p = 0.018$). There was a significant landuse × method interaction ($p < 0.001$), and a significant season × method interaction ($p = 0.009$) indicating non-covarying biodiversity dynamics between the two methods. Results for EPT also indicated non-covarying biodiversity dynamics, with significant landuse × method ($p < 0.001$), and method × season ($p < 0.001$) interactions, whereas both methods showed significant differences across seasons ($p < 0.001$) and landuse ($p = 0.004$). Chironomidae diversity showed a significant landuse × method ($p < 0.001$) interaction, as well as a significant effect of season ($p = 0.001$). Functional diversity showed significant landuse × method ($p < 0.001$) and method × season ($p < 0.001$) interactions, as well as significant seasonal ($p < 0.001$) and landuse effects ($p = 0.001$) (Table 3 and Fig. 4).

**Change in community and functional diversity over time and space.** Turnover was significantly greater than nestedness across

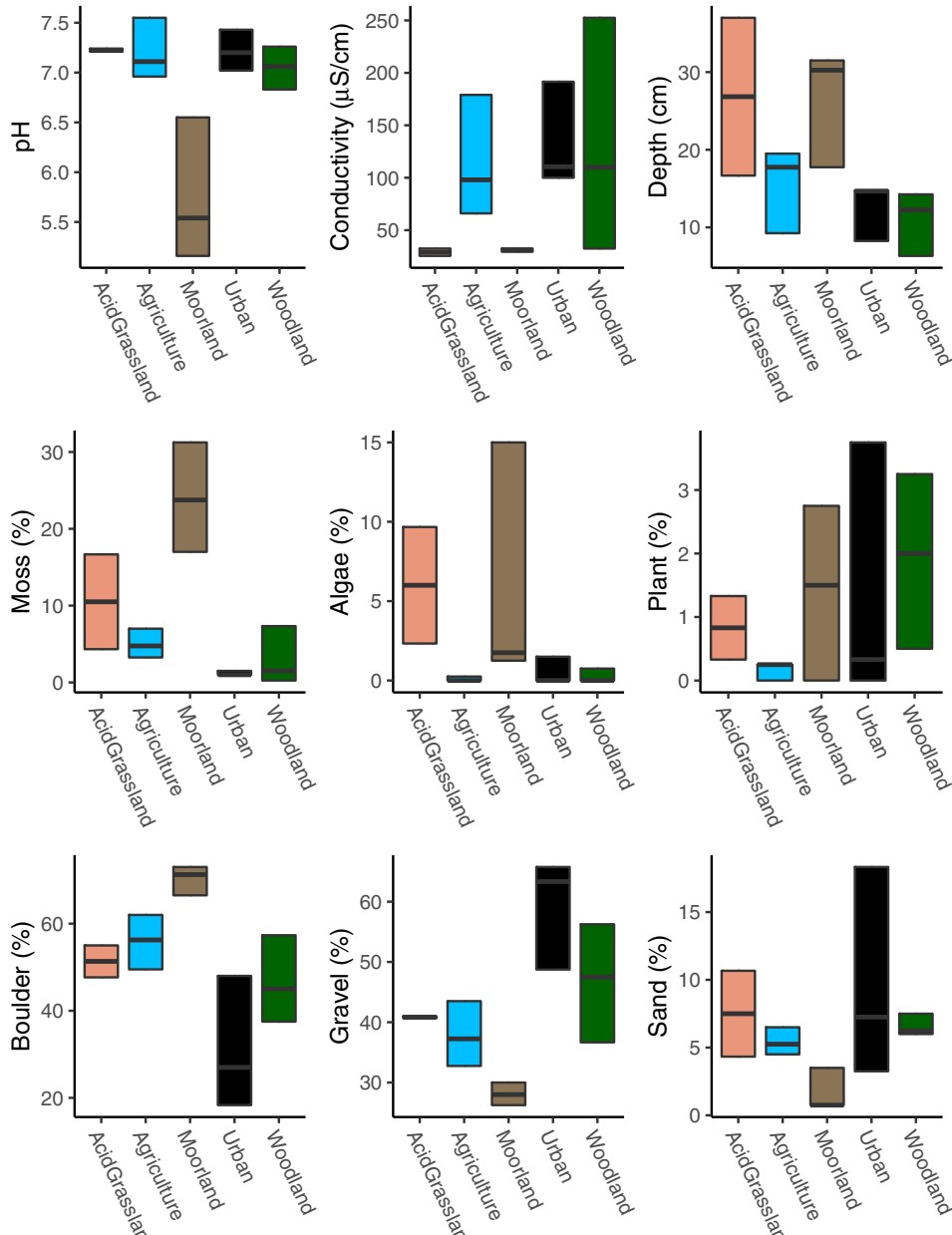

**Fig. 2 Environmental variation across landuse types.** Box plots for each environmental variable (y-axis) sampled in the study, as a single panel across sites ($N = 14$). The colors in each plot match the colors in Fig. 1 for each landuse type (x-axis). The range of the boxplots extends to the minimum and maximum of their corresponding ranges.

landuse ($p < 0.001$) and season ($p < 0.001$) for genera derived from traditional methods (Fig. 5). The variation between turnover and nestedness did not differ across season ($p = 0.660$) or landuse ($p = 0.082$), but did differ significantly between methods ($p < 0.001$), with eDNA showing greater sensitivity to detect nestedness compared to traditional methods (Fig. 5). Positive and negative relationships with regards to landuse relate to the environmental gradient described above (Supplementary Fig. 1). Changes in biodiversity over time, here turnover dominated, were directly related to the change in functionality over time (Fig. 6). For eDNA samples the increase in overall functionality was largely driven by the increase in number of genera from Diptera (all landuses), Coleoptera (Acid grasslands, Moorlands, Urban and Agriculture), Ephemeroptera (Forest), and Trichoptera

(Forest and Moorlands). Transitioning into summer, acid grasslands and moorlands showed a loss of scraper and collector functionality stemming from losses in Plecoptera and Trichoptera genera. Losses of Plecoptera genera in forest sites resulted in a loss of scraper functionality, whereas gains in Plecoptera in the urban environments showed gains in collector functionality. Agricultural sites showed gains in scraper and gatherer function, due to some increases in Diptera and Trichoptera genera. Transitioning into fall indicated loss in functionality across landuse, driven by losses in Diptera (all landuses), Coleoptera (all landuses), Ephemeroptera (agricultural, moorland), and Trichoptera (urban, forest, agricultural). Winter was largely static, with the exception of gains in functionality for agricultural sites, which was driven by increased occurrence of Plecoptera and Trichoptera

genera. Kick-net samples indicated slight increases in functionality for agricultural and moorland landuses, primarily from increases in Plecoptera (agricultural, urban) and Coleoptera (moorland) genera. Summer increases in functionality for acid grasslands and moorlands were linked to increases in Trichoptera and Plecoptera in acid grasslands, and Plecoptera and Coleoptera in moorlands. Declines in fall gatherer functionality, in agricultural and forest landuses, were driven by losses in Trichoptera

**Table 2 Number of genera recorded per sampling method by higher taxa group.**

| Higher taxa | Number of genera | |
|---|---|---|
| | eDNA | Traditional |
| Amphipoda | 1 | 1 |
| Chilopoda | 1 | 0 |
| Chironomid | 75 | 10 |
| Cladocera | 4 | 1 |
| Coelenterata | 2 | 0 |
| Coleoptera | 20 | 14 |
| Collembola | 3 | 0 |
| Copepoda | 5 | 0 |
| Ephemeroptera | 11 | 9 |
| Gastropoda | 3 | 3 |
| Hemiptera | 4 | 1 |
| Hirudinea | 4 | 3 |
| Hydracarina | 1 | 0 |
| Isopoda | 2 | 1 |
| Lepidoptera | 1 | 0 |
| Megaloptera | 1 | 1 |
| Microturbellaria | 1 | 0 |
| Nematoda | 1 | 0 |
| Neuroptera | 1 | 0 |
| Odonata | 4 | 1 |
| Oligochaeta | 23 | 2 |
| Ostracoda | 2 | 0 |
| Plecoptera | 13 | 12 |
| Porifera | 1 | 0 |
| Rotifera | 8 | 0 |
| Tardigrada | 1 | 0 |
| Trichoptera | 33 | 24 |

**Table 3 Generalized least squares (GLS) statistics.**

| Response | Explanatory | Df | F value | p-value |
|---|---|---|---|---|
| Genera | Landuse | 1 | 5.81 | **0.018** |
| | Season | 3 | 5.42 | **0.002** |
| | Method | 1 | 184.00 | **<0.001** |
| | Landuse:Method | 1 | 31.06 | **<0.001** |
| | Season:Method | 3 | 4.09 | **0.009** |
| Chironomidae | Landuse | 1 | 3.51 | 0.064 |
| | Season | 3 | 5.79 | **0.001** |
| | Method | 1 | 38.29 | **<0.001** |
| | Landuse:Method | 1 | 39.43 | **<0.001** |
| EPT | Landuse | 1 | 8.60 | **0.004** |
| | Season | 3 | 19.98 | **<0.001** |
| | Method | 1 | 422.00 | **<0.001** |
| | Landuse:Method | 1 | 13.56 | **<0.001** |
| | Season:Method | 3 | 17.04 | **<0.001** |
| Function | Landuse | 1 | 10.40 | **0.001** |
| | Season | 3 | 12.35 | **<0.001** |
| | Method | 1 | 431.53 | **<0.001** |
| | Group | 2 | 127.09 | **<0.001** |
| | Landuse:Method | 1 | 104.27 | **<0.001** |
| | Season:Method | 3 | 9.15 | **<0.001** |
| | Method:Group | 2 | 50.18 | **<0.001** |

Results include each modeled genera, Chironomidae, EPT and functional richness response variable (N = 112), tested against the set of explanatory variables; including landuse type, method (eDNA/traditional), season and the possible two-way interactions. The model includes a pairwise distance variance structure to account for spatial autocorrelation. Models shown are the most parsimonious models with the associated model metrics, including degrees of freedom (DF), f-value and p-value, given for each set of response and explanatory variables. P-values in bold indicate significant effects.

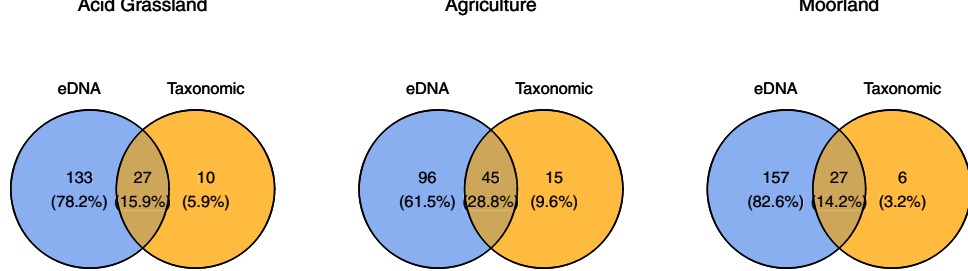

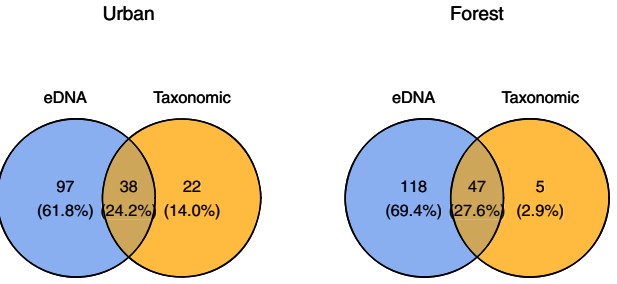

**Fig. 3 Traditional vs eDNA biodiversity similarity across landuse types.** Venn diagrams showing the proportion of genera detected for each sampling method (eDNA = blue, traditional = orange) across each landuse type. Sample sizes were 56 for eDNA (average of each sites' 3 replicate samping) and 56 for traditional sampling.

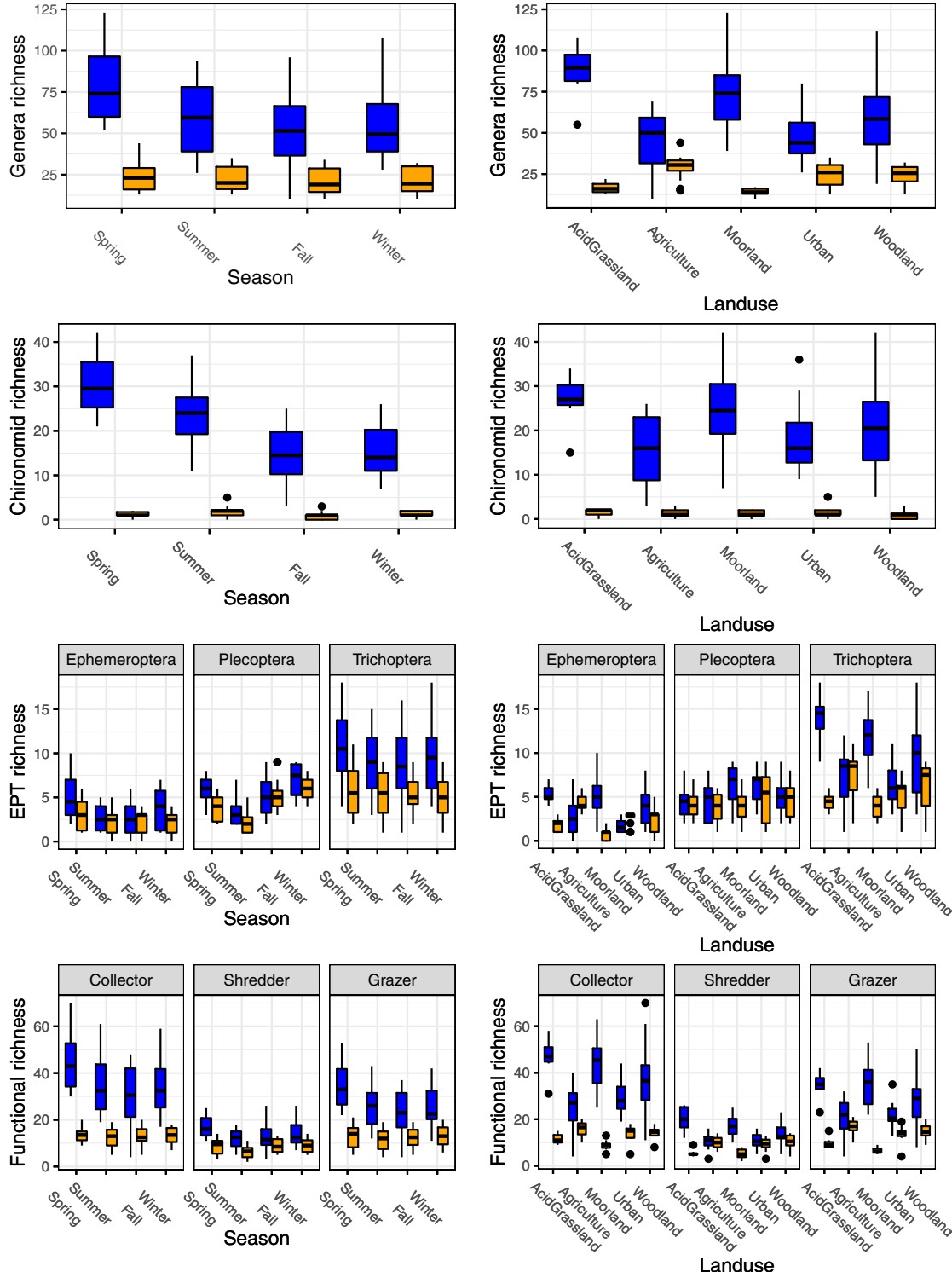

**Fig. 4 Genera, Chironomidae, EPT and functional spato-temporal richness patterns for eDNA and traditional methods.** Box plots showing the richness diversity per season (left panels), and landuse (right panels) for genera (top), Chironomidae (second from the top), EPT (second from the bottom) and functional diversity (bottom). Blue indicates eDNA derived data and orange indicates traditional method derived data. Error bars are drawn to 1.5 * inter-quartile range (IQR), with outlier points being the data outside the 1.5 * IQR range. Sample sizes were 56 for eDNA (average of each sites' 3 replicate samping) and 56 for traditional sampling.

(agriculture sites) and Plecoptera (forest sites). Loss of functionality in winter for acid grassland, agricultural and moorlands was driven by losses in Trichoptera (acid grasslands), Plecoptera (acid grasslands, moorlands), Ephemeroptera (acid grasslands, agricultural, moorlands), and Coleoptera (acid grasslands).

## Discussion

We show that eDNA based assessment offers a finer resolution of the spatial and temporal biodiversity dynamics, compared to traditional sampling. Additionally, biodiversity patterns derived from eDNA and traditional sampling showed similar general temporal

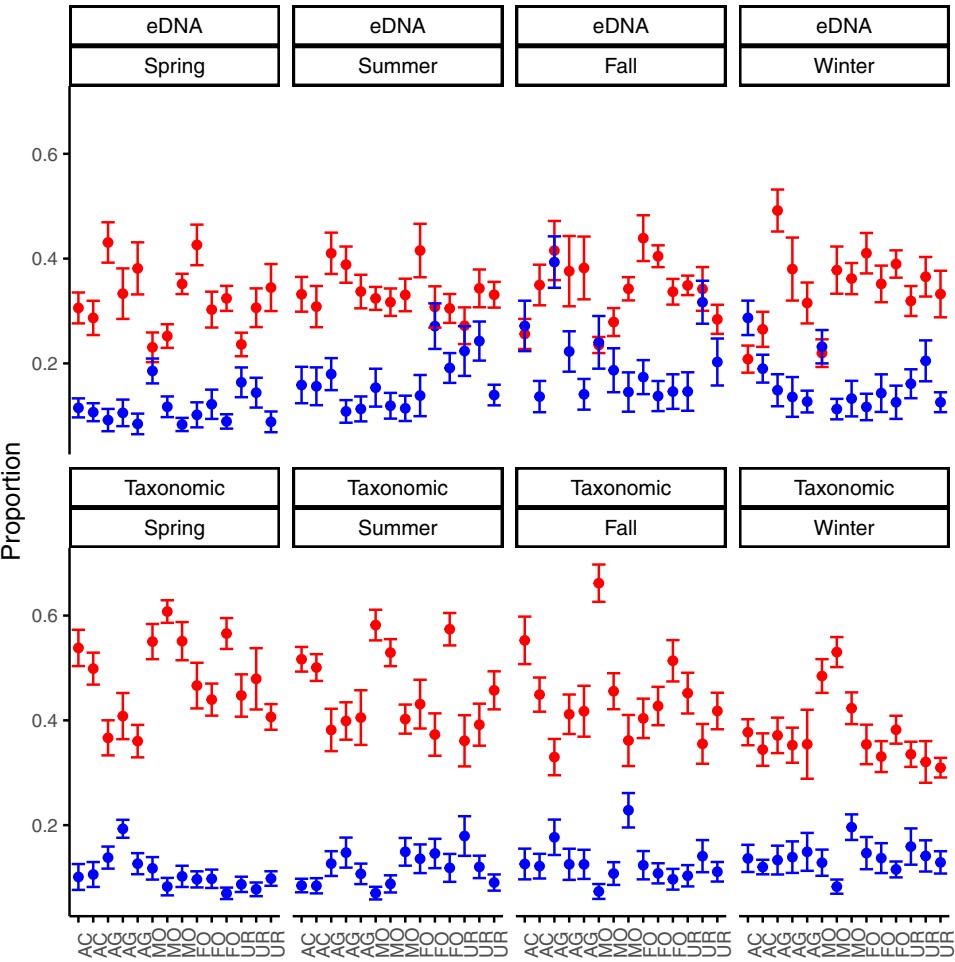

**Fig. 5 Beta-diversity partitioning of traditional and eDNA biodiversity.** Beta-diversity results partitioned into turnover (red) and nestedness (blue) components. Showing differences between methods (eDNA vs Taxonomic), season (Spring, Summer, Fall, Winter), with each site provided along the x-axis indicating its landuse type (AC = acid grassland, AG = agriculture, MO = moorland, FO = forest, UR = urban). Error bars are drawn to 1.5 * inter-quartile range (IQR). Turnover was significantly greater than nestedness across landuse and seasons ($p < 0.001$), following a gls model that accounted for spatial autocorrelation in its variance structure via a pairwise distance matrix. Sample sizes were 56 for eDNA (average of each sites' 3 replicate samping) and 56 for traditional sampling.

trends in richness and functional diversity. Localized biodiversity richness showed no environmental filtering across the landuse types, whereas partitioning of beta-diversity showed clear differences in spatio-temporal biodiversity dynamics. Specifically, regional environmental conditions were the main driver of biodiversity change, and landuse effects were less pronounced. Importantly, we show that eDNA based biodiversity assessments provide meaningful spatial and temporal relationships. Our environmental biodiversity assessment also includes increased ability to detect important indicator taxa, particularly Chironomidae and EPT taxa, which are difficult to directly sample for many of the locations, or at different time points in the year. We also show that eDNA derived functional temporal-spatial dynamics can provide clear information on how ecosystems can be effectively managed.

As expected, biodiversity was greater during spring and summer months for both sampling methods. Genera richness dynamics differed between eDNA and traditional methods among landuse types, but not among seasons where eDNA derived diversity was consistently greater than traditional diversity. Importantly, we found greater eDNA derived diversity compared to traditional methods for all sites. While several studies have shown increased observable biodiversity using eDNA over traditional methods[17,29], they have predominately focused on fish, whereas macroinvertebrate focused studies vary slightly

with most showing greater eDNA biodiversity[16,45], but also some with lower eDNA biodiversity[46], compared to traditional sampling. The disparity in macroinvertebrate diversity may stem from the increased difficulty in designing a suitable primer to capture the full range of diversity associated with macroinvertebrates, however more efficient eDNA primers are actively being developed[47]. Additionally, across landuse types, acid grasslands and moorland sites had greater eDNA diversity compared to agriculture, urban, or forest sites. The results from eDNA monitoring were more in agreement with the expectation that unmodified landuse should hold greater biological diversity, particularly with regards to the higher diversity found in moorland and acid grassland sites, which were the least modified areas in the catchment. Traditional sampling, however, suggested greater biodiversity in agricultural and urban sites, compared to moorlands or acid grasslands. The lower biodiversity observed with traditional sampling in the less disturbed sites is likely due, in part, to substrate types. The moorland and acid grassland sites are dominated by large boulders or loose sediment, which are not ideal substrates when performing traditional kick-net sampling methods, that perform better with gravel substrate (significant positive richness with increased gravel coverage $p = 0.007$), and may lead to under sampling of the local taxa[41]. Previous studies have suggested eDNA transport from upstream communities can

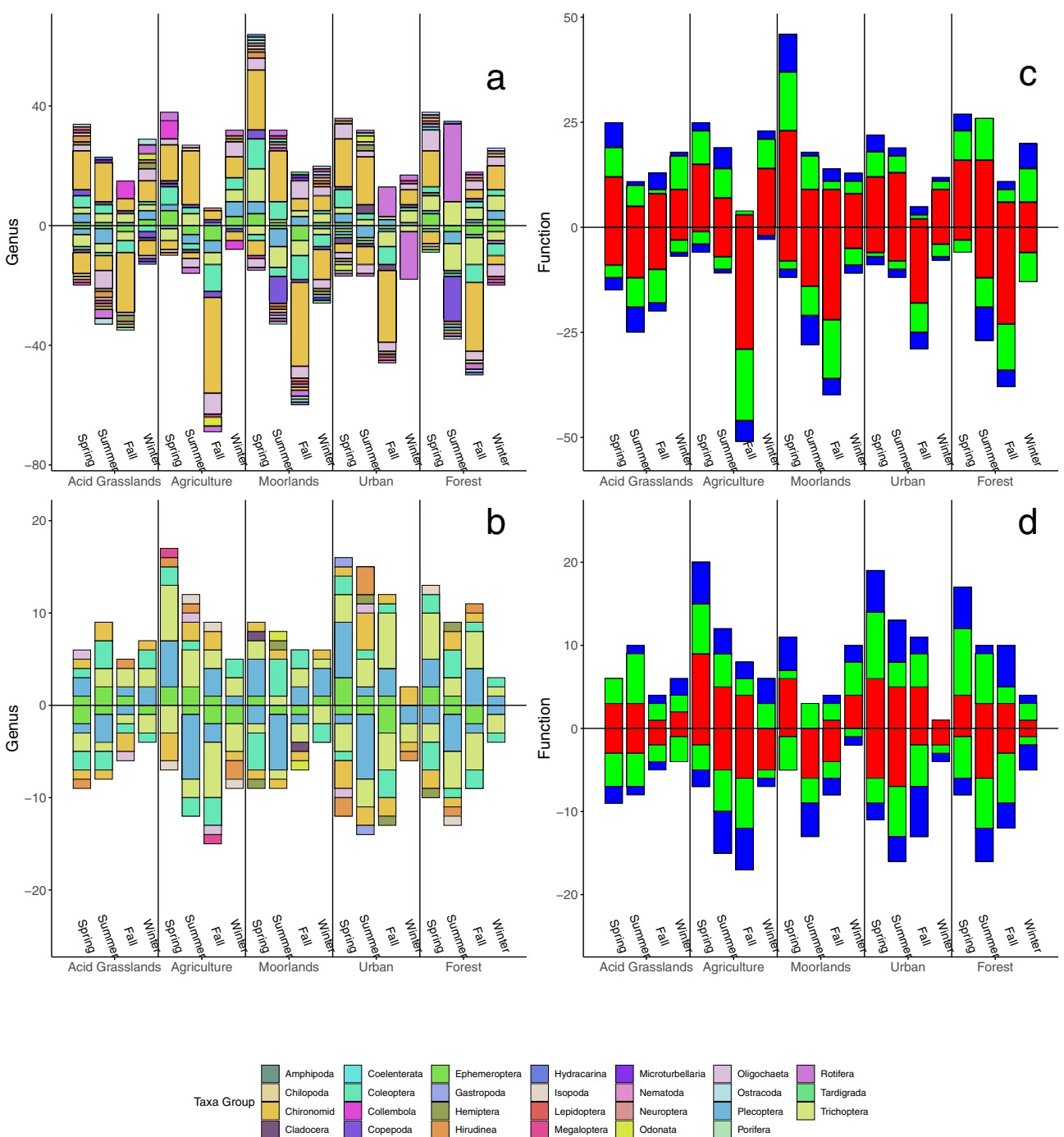

**Fig. 6 Family and functional community compositional change over time.** Change in genera richness for eDNA (**a**) and traditional sampling (**b**) and the associated change in function for eDNA (**c**) and traditional (**d**) sampling. Each subpanel is a unique landuse type showing the sequential change in genera or function whereby values above the line indicate additions in genera or function compared to the previous season and values below the horizontal line indicate loses. Colors for the genera plots are unique taxa groups, provided in the legend, and each value along the y-axis is a unique genus loss or gain. Functional group colors are: scraper = blue, collector = red, gatherer = green. Sample sizes were 56 for eDNA (average of each sites' 3 replicate samping) and 56 for traditional sampling.

increase sampled biodiversity[42,48]. There is, however, no clear indication that the landscape or catchment surrounding these sites is any more diverse, as the moorland sites in particular are at higher elevation and more isolated compared to the other landuse sites. With upstream transport limited across the study system, the increased biodiversity detection is more likely an effect of eDNA versus traditional methodology than proposed eDNA ecology. As a recommendation, eDNA sampling is likely to be

more beneficial overall compared to traditional sampling for detecting higher biological diversity, which has direct implications for detecting traditionally harder to identify biomonitoring groups such as Chironomidae or Diptera. Additionally, the use of eDNA is highly beneficial for sampling in traditionally difficult to sample locations, including non-traditional substrate types which could introduce bias when traditional, or bulk, sampling methodologies are used.

Environmental DNA biodiversity was greater for general biodiversity and all other subsets of biodiversity, including EPT, chironomids, and functional diversity (Figs. 3 and 4). The greatest increase in eDNA biodiversity resolution was in traditionally hard to identify groups, which would otherwise not be observable using taxonomic derived methods. Most revealingly, eDNA data more accurately depict Chironomidae life cycle patterns compared to traditional sampling. Specifically, seasonal variation in eDNA derived richness follows the expected larval emergence patterns, which increase over spring and summer, and steadily decline over fall and winter[28], in contrast to traditional sampling which was unable to detect this seasonal shift, possibly due to the difficulty in traditionally observing Chironomidae. Likewise, EPT emergence was detectable via both eDNA and traditional sampling, as Ephemeroptera generally emerges during the spring whilst Trichoptera emerges at more variable times throughout the year.

Spatio-temporal dynamics showed biodiversity was driven by regional turnover dynamics and less affected by localized environmental specialization or nestedness (Figs. 5 and 6). Nestedness across sites was not significant with either method, suggesting that differences in biodiversity between sites were not due to localized environmental sorting, as per our initial expectations. The primary driver of observed heterogeneity in between site biodiversity was largely due to the seasonal turnover, likely driven by the high disturbance events historically occurring in the region of the study[49], or due to very strong effects of biotic interactions[8]. There was greater eDNA turnover observed in urban and forest sites, which was attributed to higher pH and lower moss and boulder coverage compared to other sites. Turnover in EPT and Chironomidae showed similar trends compared to overall turnover, whereby the differences in biodiversity between sites were significantly attributed to the seasonal replacement of genera along the environmental gradient, which was predominately linked to pH and substrate type (Supplementary Fig. 1). Conversely, turnover was greater in moorland sites, linked to increased boulder and moss coverage. The disparity in the observed relationship between methods is likely driven largely by the methods themselves and the underlying richness values for each method, as mentioned above. Overall, both traditional and eDNA based turnover suggest seasonal turnover dominated differences in biodiversity, which could be attributed to adaptation (historical) of communities to regional conditions[50]. The biodiversity across the system is more likely a product of frequent founder and colonizing effects resulting from frequent disturbance patterns, and the inability of the sites to establish long-term interacting communities[51,52].

The functional diversity was largely dominated by collector feeders, indicating that the regional biodiversity assembly is driven by fine particulate organic matter (FPOM), which is also referred to as seston[53,54]. Two main factors contribute to the high FPOM driver across the study area. First, the widespread fecal input from animal agriculture that covers the entirety of the catchment. Additionally, for upland sites where agriculture is less prevalent, moorlands produce a high amount of FPOM[55]. The combined FPOM inputs from moorland and agricultural/urbanized environments create a regional-wide FPOM system, thereby homogenizing the functional habitat of the region as a whole. Whereas the moorland effect is likely limited to certain headwater sites in this study, the quality difference in FPOM generated from agricultural vs moorland sources likely plays a role in the diversity differences seen between sites where collectors dominate on the whole[39,53]. Seasonal shifts in functional traits were observed with eDNA, but not with traditional methods. This is reflected in the ability of eDNA to detect more functional groups compared to traditional methods, particularly with regards to Diptera, including Chironomidae and Simuliidae (i.e. blackflies), which are important collector groups that are strongly affected by

changes in temperature[55]. Likewise, eDNA functional assessment indicates a change in grazer functionality with season, closely following expectations of periphyton availability, which is a key driver of grazer activity[56]. The homogeneity in functional diversity between the landuse sites for both eDNA and traditional-based methods further indicated that environmental filtering was not the primary driver of biodiversity difference between sites. A key take-home message from assessing functional diversity in this study, over simply relying on variation in richness, is that these findings point to a clear management strategy to increase diversity across the system. Specifically, regional biodiversity would benefit by increasing the habitat for collector functional groups through improved management of local broadleaf forest, and agricultural practices to increase coarse feeding material at key headwater sites, which would increase the overall ecosystem stability of the region by increasing environmental heterogeneity.

A wider implication arising from this study is the suggestion that eDNA can disclose a much greater resolution of diversity compared to traditional approaches, and can enable multiple levels of analyses from a single data set. The benefit of comparing traditional findings with eDNA-based analyses, which can be analyzed using standardized approaches, is immensely valuable and will help avoid unintended biases introduced from cross-study traditional protocols. Overall, our findings show that eDNA is a more effective survey method to sample macroinvertebrates and provides clearer indications of the seasonal and environmental effects on multiple levels of diversity compared to traditional methods. Additionally, we provide a key assessment of regional biodiversity dynamics, which are currently underrepresented in the literature. Specifically, we show that the increased resolution of eDNA based biodiversity assessment effectively separates spatio-temporal and localized biodiversity dynamics, which here shows the importance of regional over localized management strategies. Determining such regional drivers allows for effective biological management, whereby flood control, versus altering current landuse practices, is more likely to have a greater impact on biodiversity in disturbance driven sites. Finally, by utilizing functional diversity assessment we show a clear reason for why community composition has arisen. Empowered by eDNA metabarcoding and appropriate ecological synthesis, we provide a more valuable means to describing biodiversity than simply counting unique individual units with no link to what the numbers mean at ecological and ecosystem scales.

## Methods

**Study area**. The Conwy Catchment is a 678 km² river drainage in north Wales that encompasses a wide range of habitats including forest, moorland, agriculture, light urbanization, and acid grasslands (Fig. 1). The area experiences rapid climatic shifts, particularly during winter months due to its mountainous terrain and porous rock foundations, which facilitates flash flooding, making it susceptible to periodic disturbance[49]. The Conwy Catchment area exhibits four distinct seasons that correspond to the expected life cycles of EPT and Chironomidae larval emergences throughout the year.

**Sampling**. Fourteen headwater sites across 5 landuse types (acid grassland, agriculture, forest, urban, moorland) were sampled once per season (spring, summer, fall, winter), during 2017 (5 landuse types × 14 sites × 4 seasons). Headwater sites were selected to ensure the local landuse was not influenced by other landuse types via downstream transport[48,57]. In total 168 eDNA samples and 56 traditional kick-net samples were taken over the course of the study. Sampling for each season occurred over two consecutive days. For all sampling events, streams were sampled for both eDNA and macroinvertebrate community composition during the same day. Water samples for eDNA analysis (1 L) were collected in triplicates from each stream with plastic bottles that had been cleaned prior using a 10% bleach solution (soaked for 1 h). These were then filtered through 0.22 μm Sterivex™ filter units (EMD Millipore Corporation, Billerica, USA) using a Geopump ™ Series II peristaltic pump (Geotech, Denver, USA). As filters would occasionally experience reduced filtration efficiency for different sites, or seasons, due to stream sediment loading, we would continue to run the pump for each sample until at least 500 ml was filtered, to avoid potential downstream variation in sampling[58], which we previously showed was not an issue in this experimental setup[42]. The filters were

immediately preserved in 500 µl ATL lysis buffer (Qiagen, Venlo, The Netherlands) and stored in coolers during same-day transit to the laboratory, then stored at 0 °C, for further processing. Macroinvertebrate communities were sampled using a standardized 3-minute kick-net sampling protocol, with a 500 µm mesh gauge kick-net. Kick-net sampling occurred after eDNA sampling to ensure disturbance of the site from kick-netting would not influence the eDNA signal. Both bank margins and riffle habitats were sampled during this timed sampling period. Macroinvertebrates were preserved in absolute ethanol (99.8%; VWR International, Lutterworth, UK) on collection. Upon return to the laboratory the macro-invertebrates were cleared of other collected material and identified to the lowest practical taxonomic level, per protocol[59]. Environmental (abiotic) data including pH, conductivity, depth, moss coverage, algal coverage, plant coverage boulder coverage, gravel coverage, and sand coverage, were collected for each site following UK Environment Agency site assessment protocols[59]. For each seasonal sampling event we incorporated three field blanks into the sampling process, processing one blank every 4th sampling site, resulting in a total of 12 blanks for the study. Field blanks consisted of deionized water (1 L in volume), filtered and treated the same as standard samples and were kept with the other sampling gear throughout each sampling period.

**Extraction and Sequencing.** We followed unidirectional lab practices from field, to extraction, to library preparation by using designated extraction (PCR free) and library preparation rooms. DNA was extracted from the filters using a modified QIAGEN DNA blood and tissue extraction protocol[60]. In short, 70 µl proteinase K was added directly to the filters and incubated at 58 °C overnight in a rotating hybridization chamber. Then, the lysate was extracted and the full volume was filtered through a spin column tube, after which point the standard extraction protocol was continued. Extracts (final volume 50 µl) were then cleaned for impurities using QIAGEN Power Clean kit and frozen at −20 °C for subsequent analyses. Sequencing libraries were created using a two-step protocol (see Bista et al. 2017), using matching dual end index tags (IDT) and the following COI gene region primers for the first round of PCR (PCR1): m1COIintF (5′-GGWACWG GWTGAACWGTWTAYCCYCC-3′) and jgHCO2198 (5′-TAIACYTCIGGRTGI CCRAARAAYCA-3′)[61]. Libraries were created at Bangor with the assistance of a Gilson pipette max liquid handler before being shipped to University of Birmingham's Genomic sequencing facility for quality control and sequencing. Round 1 amplification (PCR1) with the COI primers was performed in triplicates, which were then cleaned for primer dimers using magnetic beads (Beckman coulter), pooled and index labeled during Round 2 PCR step (PCR2), which were cleaned again using magnetic beads. Unique dual paired end-indices were designed and purchased from Integrated DNA Technologies, to complement the Illumina P5/P7 sequence adapters. PCR1 utilized Thermo Scientific's Ampli-gold mastermix due to the high number of inosine in the COI primer pair, and for PCR2 we utilized New England Biolab's Q5 mastermix. All PCR1 and PCR2 reactions were run in 25 µL volumes. PCR1 amplicons were generated using a reaction mix of 12.5 µL mastermix, 2 µL DNA template, 1 µL of each primer and 8 µL nuclease free water and amplified using an initial 95 °C for 5 min then 25 cycles of 95 °C for 30 s, 54 °C for 30 s and 72 °C for 60 s followed by a 72 °C final annealing for 10 min. PCR2 amplicons were generated using a reaction mix of 12.5 µL mastermix, 2 µL DNA template, 1 µL of each primer and 8 µL nuclease free water and amplified using an initial 98 °C for 30 s then 15 cycles of 98 °C for 10 s, 55 °C for 30 s and 72 °C for 30 s followed by a 72 °C final annealing for 10 min. The PCR2 amplicons were purified using High Prep PCR magnetic beads (Auto Q Biosciences) and quantified using a 200 pro plate reader (TECAN) with the Qubit dsDNA HS kit (Invitrogen). The final amplicons were pooled in equimolar quantities (at a final concentration of 12 pmol) using a Biomek FXp liquid handling robot (Beckman Coulter). Pool molarity was confirmed using a HS D1000 Tapestation ScreenTape (Agilent). Sequencing was performed on an Illumina HiSeq platform 250 bp Paired-End, with an intended coverage of 100,000 reads per sample.

**Bioinformatics.** Bioinformatic processing up to taxonomic assignment was performed by University of Birmingham. In short, per base quality trimming was performed on demultiplexed reads using SolexaQA+ +v.3.1.7.1 (Cox, Peterson, & Biggs, 2010) and paired end reads were merged using Flash v.1.2.11[62], with default parameters. Primer sequences were removed with TagCleaner v.0.16[63] allowing up to 3 mismatches per primer sequence. Only sequences with both forward and reverse primers were retained for further analyses. Amplicon sequence variants (ASVs) were obtained via Usearch at 97% similarity threshold, and denoising with the –unoise3 algorithm. Chimeras were removed as part of the –unoise3 algorithm[64]. Taxonomy to the genus level was assigned to representative ASV with BLAST against ASVs using the non-redundant nucleotide database of NCBI, using the default settings[65].

**Metacommunity analyses.** All statistical analyses were performed using R version 3.6.1[66]. Sequence reads were rarified for each set of replicates to the lowest replicate level. Mean number of reads for each ASV were calculated across the sample replicates before being matched to their taxonomic identifier. ASVs that were not identifiable to the genus level, or to a functional group (below), were not included in subsequent analyses. Genera richness was calculated as the number of unique

genera per site. We further divided richness into unique EPT genera for EPT richness, and Chironomidae richness as the number of unique Chironomidae genera. Functional richness was calculated as the partition of unique functional groups per sample, following the partition of functional groups in Moog (2017). In short, Moog (2017) provides a catalog of 3296 metazoan species that form the basis of ecological status assessment for many European environmental agencies. The functional scores are assigned to each taxa based on a ten point partitioning to reflect the variation in functionality within taxa, meaning the functional scores is a score of the function and not simply a reassignment of the taxonomic identification. These groups reflect the functional feeding groups, divided into 8 categories, including; grazer/scrapers, xylophagous, shredders, gatherers/collectors, active filter feeders, passive filter feeders, predators, and other. We refined these groups to shredders, grazers, and collectors whereby collectors were the summation of gather/collectors and filter-feeding groups to simplify the functional groups to those used across wider studies.

**Statistics and reproducibility.** To reiterate, the final dataset used for the final analyses included 168 eDNA samples and 56 traditional kick-net samples. Environmental DNA sampling consisted of three replicates per sample, whereas traditional sampling involved single unit sampling. Environmental data sampling included one set of sampling per seasonal sampling, averaged across the seasons. To assess community dynamics between sites we calculated the nestedness and turnover components of beta-diversity following Baselga (2010)[14], whereby beta-diversity was calculated as $\beta sor$ (1), Turnover as $\beta sim$ (2) and $\beta nes$ as $\beta sor$ minus $\beta sim$ (3)[14]. All mathematical formula follow the nomenclature of Baselga (2010), with $a$ being the number of genera common to both sites, $b$ is the number of genera occurring in the first site but not the second and $c$ is the number of genera occurring in the second site but not the first.

$$\beta sor = \frac{b + c}{2a + b + c} \tag{1}$$

$$\beta sim = \frac{\min(b, c)}{a + \min(b, c)} \tag{2}$$

$$\beta nes = \beta sor - \beta sim \tag{3}$$

We used generalized least squares (gls), as implemented using the gls function in the nlme package[67], to assess the statistical relationships between community, EPT, Chironomidae, and functional richness (each as a separate response variable and independent statistical test) and all two-way interactions of the explanatory variables, including sampling method (eDNA or traditional), season (spring, summer, fall, and winter) and landuse gradient (see below for description). Generalized least squares is an extension of linear regression that allows for variance structuring of variables to account for suspected correlation between residuals. Here, we specifically used the gls framework to account for potential spatial autocorrelation between sites by including a variance structure using the distance matrix for the sampling sites[67]. We further used gls to assess the statistical relationships between nestedness and turnover between communities against a set of explanatory variables, including the landuse gradient, sampling method, and season, including all possible two-way interactions. Backward model selection was performed to find the most parsimonious model using Akaike information criterion (AIC) to determine the best model fit[68]. Model assumptions, including normality and heteroscedasticity using model diagnostic plots were implemented in R. The landscape gradient was calculated as the first principal component of a PCA derived from non-covarying environmental variables (normalized and centered prior to PCA analysis), where non-covariance was assessed via visualization of the pairwise correlation between all measured environmental variables (Supplementary Fig. 1)[68]. The first axes of the PCA, which was used as the derived environmental gradient, accounted for 67.47% of the observed variation with environmental loadings consisting of pH (0.027), moss coverage (−0.449), depth (−0.297), and boulder coverage (−0.842).

**Reporting summary.** Further information on research design is available in the Nature Research Reporting Summary linked to this article.

## Data availability

All source data underlying the graphs and charts presented in the main figures are available via FigShare https://doi.org/10.6084/m9.figshare.14159579.v1[69].

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

## Acknowledgements

Centre for Ecology and Hydrology for field surveying, sampling assistance, and macroinvertebrate identification. Genomics Center at University of Birmingham for sequencing and initial bioinformatics services. The project was supported by a Natural Environment Research Council grant (parent-grant NE/N006216/1; sub-grants NE/N005724/1 and NE/N00576/1).

## Author contributions

Designed Research: M.S., F.K.E., B.J.C.; Performed Research: M.S., P.M.S., F.K.E., I.B.; Analyzed Data: MS; Wrote Paper: M.S.; Contributed critically to initial drafts of the paper: F.K.E., B.J.C., I.B., G.R.C., M.dB. All authors, including S.C., F.L.B., and H.C.G. commented on the final version of the manuscript.

## Funding

## Competing interests

The authors declare no competing interests.
