## [Peer Review File · Communications Biology]

Reviewers' comments:

Reviewer #1 (Remarks to the Author):

Seymour et al. used eDNA and traditional kick-net sampling to investigate how the alpha, beta and functional diversity of macroinvertebrates change across seasons and landuse types. To achieve this goal, they sampled 14 sites in 2017, with one sampling event per season (spring, summer, fall, winter) across 5 landuse types. This sampling campaign generated a total of 168 eDNA and 56 kick-net samples. The eDNA samples were amplified by targeting the COI gene and pair-end sequenced using Illumina HiSeq technology while the kick-net samples were morphologically identified to the lowest possible taxonomic level. They found that eDNA derived genera richness was significantly higher than traditional kick-net derived richness across seasons and landuse types. There was no significant effect of landuse type on both eDNA and traditional kick-net derived genera richness. They conclude that eDNA is an effective method to assess temporal and spatial patterns of macroinvertebrate diversity, and that this technique can be used to design effective biodiversity assessment and management strategies.

The idea behind the paper is great and worth investigating. The objectives of the paper, though incomplete, are clear and the hypotheses are testable. The sampling design is perfect and the number of samples collected is impressive. My main worries with this manuscript are related to the way the analyses were done and the way the results are presented. These need to be carefully addressed in order to improve the overall quality of the manuscript.

Specific comments

1. Lines 82-86: This statement is true but please provide relevant references to back up this claim, i.e. studies showing that eDNA is more sensitive than traditional surveys [1-5]. Please also note that this is not always true in all individual cases (reviewed in Beng & Corlett 2020 [6]). There are cases where traditional surveys outperform eDNA and where both traditional surveys and eDNA show similar performance [7-9]. In addition to the taxa- or species-specific differences in sensitivity between eDNA and traditional surveys, the environment, time of the year, and biotic factors also play important roles [10-16]. In aquatic ecosystems, for example, eDNA can persist from a few hours to a month after release [10, 11]. Differences in eDNA persistence can occur even within the same environment, for example, between the surface and bottom layers of a water body [12, 13, 16].
2. Lines 97-99: The objective "We investigate whether metabarcoding of headwater river eDNA is effective for the assessment of macroinvertebrate community diversity dynamics across season and landuse types" is clearly stated but this was not the sole objective of the study. Please list all objectives sequentially and perhaps match each objective with the 3 hypothesis below.
3. Hypothesis 1: With regards to seasonal communities of riverine macroinvertebrates, richness or alpha diversity, is expected to peak during spring and summer, when many stream macroinvertebrates are emerging as adults and reproducing, compared to fall and winter months when total biomass of many species has declined (Lines 99-103).
4. Hypothesis 2: Environmental filtering effects that are linked to habitat modification, such as agriculture or urbanized areas, are expected to negatively impact macroinvertebrate diversity, particularly Chironomidae and Ephemeroptera, Plecoptera and Trichoptera (EPT) taxa, and lead to homogenized functional diversity through the combined processes of environmental filtering and biotic interactions (Lines 103-107).
5. Hypothesis 3: Utilizing the nested and turnover components of inter-community similarity (i.e. beta-diversity), we can expect high turnover within sites as community assembly changes over time, and high nestedness across environmental sites, attributed to environmental filtering of the localized sites. Alternatively, low nestedness could indicate a low effect of environmental filtering

and a greater effect of stochastic or biotic factors influencing the localized community assemblies (Lines 107-112).

6. The sampling scheme was well designed and the number of samples (168 eDNA and 56 traditional kick-net) collected is impressive. Thank you for investing so much time and effort to generate such a comprehensive dataset.

7. Lines 191-193: Usearch has an algorithm (UCHIME) for denovo chimera detection [17]. I am curious to know if chimera detection/removal was performed in this study and if not, how were chimeric sequences detected and removed.

8. Lines 193-194: Please include a reference for NCBI-BLAST [18]. I assume that default settings (e.g. Max target sequences = 100, Expected threshold = 0.05, Word size = 28, etc) were used but please mention it here, and if not please state any changes or modifications.

9. What was the total number of OTUs obtained in this study and were OTUs present in only one sample (singletons) or in a few samples removed prior to downstream analyses? It will be interesting to also report the total number of species detected using kick-net.

10. Lines 195-200: What eventually happened to these 676 contaminant reads? It could happen that contamination in the negative controls is also present in the actual samples. In this case, all OTUs shared between negative controls and actual samples should be discarded prior to downstream analyses.

11. What is the difference between the sections 'Metacomunity diversity measures' (Line 202) and 'Statistics' (Line 218)? Both contain information on how 'data' was analyzed. To improve readability and reduce complexity, I would merge these as a single section and call it 'Data analysis'.

12. Lines 204-205: What was the 'lowest replicate level', 10,000, 20,000 or 30,000 reads? Please provide a value so that readers can have an overview of the quality and quantity of data being analyzed.

13. Lines 205-206: I am sorry but I don't understand the reason behind this sort of analysis. Why was this necessary and is the result from this analysis available somewhere? If not please provide this result in the supplementary material.

14. Lines 206-209: In most cases, species richness is used as a metric to measure alpha diversity. Here, genera richness was used. It would be nice to at least state why genera instead of species richness were used. A possible explanation could be that genera richness is a surrogate of species richness and that this has been shown for many taxa [19-23].

15. Lines 209-215: The definition of functional richness here is too simple and the overall results are not at all different from genera richness (Fig 4). It's like doing the same analysis but calling them different names. I recommend looking at functional richness from a more robust perspective [24, 25] and using metrics that are not highly dependent on taxonomic richness. The following R packages might be useful. (i) *cati* (<https://cran.r-project.org/web/packages/cati/cati.pdf>), (ii) *funrar* (<https://cran.r-project.org/web/packages/funrar/funrar.pdf>), (iii) *FD* (<https://cran.r-project.org/web/packages/FD/FD.pdf>). But feel free to explore other alternatives.

16. Lines 215-216: The citation Baselga (2010) is missing from the list of references [26]. Stating that "Beta diversity was calculated as nestedness and turnover components following Baselga (2010)" is not enough. Please provide details of the functions used (e.g. 'beta-multi.R', 'beta-pairwise.R' or both), state whether resampling was involved or not and how many random samples were used.

17. Line 221: What is 'community'? I guess this refers to 'Genera richness' as presented in the tables and figures. If yes, then please keep it consistent throughout the manuscript. Either use 'community' or 'Genera richness' throughout but not both. In each case, a clear definition from the beginning would be useful. e.g. Community refers to the total number of genera in the whole dataset. The same for 'landuse gradient' (line 227) and 'landscape gradient' (line 228). Using different terms to refer to the same thing is confusing to the readers.

18. Line 223: How about using 'eDNA or kicknet' instead of 'eDNA or traditional'? The former reminds readers of the specific techniques used whereas 'traditional' could mean 'anything'.
19. Line 228: 'first principal ...' or 'first principle ...'? It is not clear why the first principal component of a PCA was used in linear regression instead of using the individual non-covarying environmental variables. If I remember correctly, the samples were collected from 5 different landuse types (urban, agriculture, acid grassland, moorland, and forest), and these already represent a 'gradient'. Besides, there is no mention of how well PC1 captured this so-called 'landuse/landscape gradient' i.e. how much of the variation in the dataset was captured by PC1? Which environmental variables were highly loaded on PC1? Did PC2 also explain a considerable proportion of the variance? I would present these PCA results in the 'Results' section instead of presenting Table 1 and Figure 2. They contain valuable information that could be potentially used to interpret the results of the linear regression.
20. Lines 241-242: What is the importance of providing summary statistics of environmental variables (Table 1 and Figure 2) that were never used in any of your analyses and which do not directly address the objectives of this study? Since these individual variables were reduced using PCA, presenting results of the PCA makes more sense as readers can use these PCA results to better understand or interpret the linear regression results.
21. Lines 241-243: The presentation style of these results needs to be reconsidered. In the 'Results' section, authors are supposed to first summarize 'what they found' before referring readers to Tables and/or Figures (if any). It is strange to see statements like "All environmental variables and their associated summary statistics are presented in Table 1 & Figure 2. Variation in genera alpha, Chironomidae and EPT richness is presented in Figure 3. The full breakdown of genera per landuse type can be found in supplementary table S1" without first summarizing the results. For instance, a summary statement such as "On average, pH was relatively lower in moorland while Gravel was relatively higher in Urban than in other landuse types (Fig 2). Or "Moorland had the lowest mean pH, Gravel and Sand, and the highest Depth, Moss and Boulder compared to the other landuse types (Fig 2). Besides, if Fig 2 must be presented, please perform an ANOVA test to compare the mean of the environmental variables across landuse types. If there's a significant effect of landuse, please proceed to perform multiple pairwise comparisons to determine if the mean between specific pairs of landuse (e.g. Acid grassland vs Agriculture) are statistically significant (<http://www.sthda.com/english/wiki/one-way-anova-test-in-r>). You can do the same for Figure 3. Simply presenting boxplots is not enough given that readers need to know if the observed differences (genera, Chironomidae, EPT, and functional diversity) are statistically significant.
22. The authors presented some interesting hypotheses at the end of the Introduction (lines 99-112) but these have not been satisfactorily tested. To test these hypotheses, ANOVA and multiple pairwise comparisons are inevitable (See the link in the previous comment on how to perform such analyses).
23. Lines 246-247: It is simply reported that the Moran's I test for spatial autocorrelation was non-significant or significant but no further details are provided on how the 'significant' Moran's I (which signifies strong spatial autocorrelation) was dealt with. Normally, this 'significant' Moran's I should be included in linear regression to account for spatial autocorrelation [27-29] but it looks like this was not the case here. I wonder why it was computed in the first place.
24. Lines 249-270: The section heading "Method comparison" is misleading. Which methods are being compared here? The contents of this section are the results of linear regression and naming it "Drivers of taxonomic and functional richness" would be reasonable. Spatial autocorrelation should be accounted for in these analyses and if possible use the individual non-covarying environmental variables for easy interpretation. Please make a clear distinction between the 'Landuse' in Table 2 and in Figure 3. It's worth emphasizing in the title that the 'Landuse' in Table 2 represents or was

obtained from PCA.

25. Lines 250-251: I guess there is a typo in this sentence "Genera richness derived from eDNA was significantly greater than traditional derived eDNA across season ($p < 0.01$) and landuse type ($p < 0.001$). The second 'eDNA' should be 'richness'? In short, please rephrase.

26. Lines 261-270: Judging from the title of the manuscript, these are the key findings. Here, it would be good to first present a figure showing the partitioning of beta diversity into turnover and nestedness components for eDNA and kicknet together. Similar graphs for genera, Chironomidae, EPT, and functional richness would be a plus. Perhaps Figure 2 of Baselga & Orme [30] can be of help.

27. Line 272: These analyses could be improved further using ANOVA and multiple pairwise comparisons tests. See previous comments above.

28. Lines 71-72: This sentence should end with a full stop (.) and not a comma (,)

29. Lines 73-77: These sentences are currently difficult to read, please consider rephrasing them. "Taxonomic identification of freshwater macroinvertebrates for biomonitoring is largely limited to mature life stages that (and) can be (difficult) different among taxonomically similar species or genera. The high level of taxonomic specialization required to identify specimens and the long processing times per traditional sample (rendering) render large scale ecosystem-wide traditional assessments expensive and time consuming"

References

1. Kraaijeveld, K., et al., Efficient and sensitive identification and quantification of airborne pollen using next-generation DNA sequencing. *Molecular Ecology Resources*, 2015. 15(1): p. 8-16.
2. Olds, B.P., et al., Estimating species richness using environmental DNA. *Ecology and Evolution*, 2016. 6(12): p. 4214-4226.
3. Deiner, K., et al., Environmental DNA reveals that rivers are conveyor belts of biodiversity information. *Nature Communications*, 2016. 7: p. 12544.
4. Tingley, R., et al., Environmental DNA sampling as a surveillance tool for cane toad *Rhinella marina* introductions on offshore islands. *Biological Invasions*, 2019. 21(1): p. 1-6.
5. Strickland, G.J. and J.H. Roberts, Utility of eDNA and occupancy models for monitoring an endangered fish across diverse riverine habitats. *Hydrobiologia*, 2019. 826(1): p. 129-144.
6. Beng, K.C. and R.T. Corlett, Applications of environmental DNA (eDNA) in ecology and conservation: opportunities, challenges and prospects. *Biodiversity and Conservation*, 2020. 29(7): p. 2089-2121.
7. Hanfling, B., et al., Environmental DNA metabarcoding of lake fish communities reflects long-term data from established survey methods. *Molecular Ecology*, 2016. 25(13): p. 3101-3119.
8. Yamamoto, S., et al., Environmental DNA metabarcoding reveals local fish communities in a species-rich coastal sea. *Scientific Reports*, 2017. 7.
9. Hopken, M.W., et al., Molecular forensics in avian conservation: a DNA-based approach for identifying mammalian predators of ground-nesting birds and eggs. *BMC Research Notes*, 2016. 9(1): p. 14.
10. Dejean, T., et al., Persistence of Environmental DNA in Freshwater Ecosystems. *Plos One*, 2011. 6(8).
11. Pilliod, D.S., et al., Factors influencing detection of eDNA from a stream-dwelling amphibian. *Molecular Ecology Resources*, 2014. 14(1): p. 109-116.
12. Lacoursière-Roussel, A., et al., eDNA metabarcoding as a new surveillance approach for coastal Arctic biodiversity. *Ecology and evolution*, 2018. 8(16): p. 7763-7777.
13. O'Donnell, J.L., et al., Spatial distribution of environmental DNA in a nearshore marine habitat. *PeerJ*, 2017. 5.
14. Barnes, M.A. and C.R. Turner, The ecology of environmental DNA and implications for

- conservation genetics. *Conservation Genetics*, 2016. 17(1): p. 1-17.
15. Takeuchi, A., et al., Release of eDNA by different life history stages and during spawning activities of laboratory-reared Japanese eels for interpretation of oceanic survey data. *Scientific Reports*, 2019. 9(1): p. 6074.
 16. Anglès d'Auriac, M.B., et al., Detection of an invasive aquatic plant in natural water bodies using environmental DNA. *PLOS ONE*, 2019. 14(7): p. e0219700.
 17. Edgar, R.C., et al., UCHIME improves sensitivity and speed of chimera detection. *Bioinformatics (Oxford, England)*, 2011. 27(16): p. 2194-2200.
 18. Altschul, S.F., et al., Basic local alignment search tool. *Journal of Molecular Biology*, 1990. 215(3): p. 403-410.
 19. Andersen, A.N., Measuring more of biodiversity: Genus richness as a surrogate for species richness in Australian ant faunas. *Biological Conservation*, 1995. 73(1): p. 39-43.
 20. Balmford, A., A.H.M. Jayasuriya, and M.J.B. Green, Using Higher-Taxon Richness as a Surrogate for Species Richness: II. Local Applications. *Proceedings: Biological Sciences*, 1996. 263(1376): p. 1571-1575.
 21. Báldi, A., Using higher taxa as surrogates of species richness: a study based on 3700 Coleoptera, Diptera, and Acari species in Central-Hungarian reserves. *Basic and Applied Ecology*, 2003. 4(6): p. 589-593.
 22. Alves, C., et al., Genera as surrogates of bryophyte species richness and composition. *Ecological Indicators*, 2016. 63: p. 82-88.
 23. Balmford, A., M.J.B. Green, and M.G. Murray, Using higher-taxon richness as a surrogate for species richness: I. Regional tests. *Proceedings of the Royal Society of London. Series B: Biological Sciences*, 1996. 263(1375): p. 1267-1274.
 24. Laureto, L.M.O., M.V. Cianciaruso, and D.S.M. Samia, Functional diversity: an overview of its history and applicability. *Natureza & Conservação*, 2015. 13(2): p. 112-116.
 25. Hatfield, J.H., M.L.K. Harrison, and C. Banks-Leite, Functional Diversity Metrics: How They Are Affected by Landscape Change and How They Represent Ecosystem Functioning in the Tropics. *Current Landscape Ecology Reports*, 2018. 3(2): p. 35-42.
 26. Baselga, A., Partitioning the turnover and nestedness components of beta diversity. *Global Ecology and Biogeography*, 2010. 19(1): p. 134-143.
 27. Bispo, P.d.C., et al., Drivers of metacommunity structure diverge for common and rare Amazonian tree species. *PLOS ONE*, 2017. 12(11): p. e0188300.
 28. Liu, J.-J. and J.W.F. Slik, Forest fragment spatial distribution matters for tropical tree conservation. *Biological Conservation*, 2014. 171: p. 99-106.
 29. Slik, J.W.F., et al., Large trees drive forest aboveground biomass variation in moist lowland forests across the tropics. *Global Ecology and Biogeography*, 2013. 22(12): p. 1261-1271.
 30. Baselga, A. and C.D.L. Orme, betapart: an R package for the study of beta diversity. *Methods in Ecology and Evolution*, 2012. 3(5): p. 808-812.

Reviewer #2 (Remarks to the Author):

Review of Seymour et al – Nestedness and turnover of riverine species and functional diversity using eDNA and traditional approaches.

This manuscript describes the use of eDNA and traditional sampling methods to assess macroinvertebrate community diversity. In addition the effects of land use and seasonality on community and functional diversity is assessed. This is a very thorough study with interesting findings which will strengthen knowledge in this area and could be used to inform management

strategies.

I think that the statistical analysis presented is of sound quality and appropriate to this study. A small amount of additional detail in the methods section (see below) will further make this study reproducible.

I have a number of comments to make many of which are minor:

1. In the title I don't think you can use 'species' as you are discussing genus/order/family level changes. You could use 'macroinvertebrates'. I think you may also need to be clear as to the level to which you are referring in the abstract as initially I thought you would be discussing results down to the species level.
2. Full stop needed at end of line 72.
3. Line 76 'rendering' should this be 'renders'?
4. Lines 82-86, very long sentence, is it possible to split into two or three sentences.
5. Line 88 – reference examples of this.
6. Line 93 – again any references for this statement?
7. Is an ethics statement required for destructive sampling?
8. Figure 1 Legend – delete 'Colors)', 'landuse' is two separate words (throughout document), and I'd add the number of sampling locations.
9. Line 131 - I assume eDNA was collected first at each site, specify this.
10. Line 132 – was there any variation in the amount of water sampled i.e. did 1L always go through the filter okay?
11. Line 134 – how long were bottles soaked in 10% bleach?
12. Line 136 – how much ATL buffer was used?
13. Line 138 – reference for standardised protocol?
14. Line 141-143 – how, microscopy, by eye, were any identification keys used (if so reference these).
15. Line 143-146 – when taken before/after eDNA? I'm asking to ensure that there was as little disturbance of the site prior to eDNA sampling as possible.
16. Line 146-150 – this is not entirely clear as you have not specified how many sampling events there were in total and you only say three field blanks yet there are 12 blanks in total, if three field blanks sampled in triplicate or three per sampling event, this would only be 9 samples. What volume are the field blanks? I think a little bit of clarification is required here. Were there any laboratory blanks or is this what you are referring to from 148 on?
17. Line 152 – was a unidirectional lab flow followed, you should specify this.
18. Line 155 – volume of PK and temperature?
19. Line 158 – what was the final volume/concentration of the extracted DNA, I would add this information?
20. Line 167 – name of indexing kit used?
21. Line 168 – remove capital 'D' from 'Due'.
22. Line 176 – were the PCR1 amplicons purified prior to PCR2? Picky, I know but what make and model of PCR machines were used, I think that all of this needs to be included?
23. Line 178 – used Qubit twice
24. Line 193 – was assignment to species or family level?
25. Lines 195-200 – these are results so should not be in the methods section. I also think that the results section would benefit from some data output information in tabular form although this might be better in the supporting information.
26. Line 213 – I think you should specify which families/genus are in each of these groups.
27. Line 215-216 – might be useful to put in a brief description of beta diversity calculation (or in supplemental information).

28. Figure 3 – I think that the right side panels need to be the same size as the left side panels. Also need to add in what the dots refer to presumably outliers and if so an explanation as to why excluded.
29. Table S1 – I think it would be useful to include percentage totals per family/site and method and also include unassigned percentages too. I know you have split into two separate tables and this looks good, but I can't help but think it would be easier to compare the two methods if the two methods were in one table. This is obviously a personal preference, but I was interested to see how different the results were between the two methods and there is no easy way to do this than to scroll back and forth. I do think that although not the focus of the paper that you should include more discussion on how comparable the two methods were e.g. any gaps in the eDNA data and/or representative venn diagrams (could add in around lines (302-304 or 316).
30. Figure 4 – for panels A and B, what do the different colours represent – needs a key of the different genera so that the text lines 273-294 has some context. Where is the data for the previous season as referred to in the figure legend? I think this figure also needs more explanation, I understand that functional groups can show losses and gains as some families within e.g. scrapers will be lost and some gained compared to the previous season, for the genus graphs this is the case too but I think you need to spell this out by giving an example e.g. within the genus shown in green X and Y showed gains and Z showed a loss.
31. Lines 273-294 – I would highlight any obvious differences between the eDNA and traditional data within the results section too.
32. Line 307 'timepoints' is two words.
33. Line 316-319 – I would add in reference to the results figures again in the discussion to make it easy for the reader to cross check.
34. In line 331 you suggest why there may be greater macroinvertebrate diversity at moorland sites which sounds reasonable to me but you also explain why there is lower diversity with traditional methods in moorland sites (lines 323-326), I find this contradictory so this may need to be edited to make clear.
35. Line 345 – I would use 'Secondly' rather than 'And second' and also 'Firstly' in line above. Sentence is also very long and doesn't totally make sense, is there one too many 'and's i.e. 'moorland sites, the' rather than 'moorland sites and the'?
36. Lines 386-391 – I would start a new sentence at the semi-colon in line 388.
37. Line 393-394 – how does this compare to other studies that have compared traditional and metabarcoding approaches even if on different environments/taxa? I think this requires some discussion.

--Signed Helen C. Rees

Reviewer #3 (Remarks to the Author):

Review

The paper entitled "Nestedness and turnover of riverine species and functional diversity using eDNA and traditional approaches" by Seymour et al. presents a robust dataset comparing traditional field sample against eDNA methods for macroinvertebrates. The manuscript presents a detailed description of the ecological findings and meaning for ecological monitoring.

Comments

1. Comparing methods. It is not surprising to find that eDNA methods provide much higher diversity

values. eDNA methods are known to be much more sensitive, or at least not that surprising. This point is important to show, particularly for benthic macroinvertebrates which are used for water quality monitoring. One large concern here is that any analysis comparing the methods is a result of this sensitivity difference and potentially the fact that eDNA samples (although 1L) represent a much large area of sampling than the field methods. The sampled water is an integration of water upstream. For the monitoring story line, it seems more important to also communicate any genera that were caught in the field sample but not in the eDNA method. Also, it seems important to mention the scale of inference between the two methods (see inference section in Goldberg et al 2016). The introduction does not overly state this point. The Results section on this does seem overly detailed and statistical, rather than really comparing how the methods comparison influences monitoring – method replacement or combined approach. A deeper discussion into the meaning of comparing the methods, might change the Discussion paragraph starting on Line 312 and 341. To be clear, I think it is debatable that there is utility in comparing data from these methods. However, for this dataset and manuscript, I think this point should be made clear, so the reader understands the issue before dropping into the results and discussion. The field data could be presented as validation not comparison.

2. Unclear storyline. The ecological story of seasonal diversity change is interesting and confirming presumed patterns with better data. The monitoring story is a clear and very important one for the applied implications. However, I think the manuscript is missing a clear ‘game changer’ story line. See this recent paper for an example of a very similar topic:

<https://www.nature.com/articles/s41467-020-17337-8> I think there are two about two separate topics (monitoring and diversity). The conclusions are clear in the manuscript but not perhaps presented for a general audience. For a general audience, not a community ecology one, the manuscript’s conclusion feels overly contextual and disciplinary. The monitoring story, if developed, could lead to significant attention (just my opinion). Given the dataset, I think a lot can be learned from this work. I look forward to more papers.

Smaller Points:

1. Line 312 – the point above that the methods might not be appropriate to compare.
2. Two storylines – I am struggling a bit with understanding the main message for the paper. There is a lot here and this dataset is impressive. The two storylines to me are: using of eDNA compared to traditional methods for BMI; (2) biodiversity (all of them) in stream BMI. The two are of course intertwined. I feel the manuscript could be improved by a better explanation why diversity is important to understand in environmental monitoring. Potentially, there are two papers here one on environmental monitoring using eDNA and one on the basic ecology.
3. Line 50 – Implies that eDNA methods might be more cost effective. I do not think this is true yet, nor have I seen data on it other than our own. This is a larger issue that probably needs to be danced around a bit or better explained.
4. Figure 1. Sites were classed by the different land uses, yet it is unclear from Figure and the text how the land use classification was made for mixed land use sampling sites. This is important for riverine biodiversity locally since it is driven upstream source characteristics. Figure 1 could be improved by showing the sub-watershed boundaries and streams. Adding the land use classed for each site would also help the reader better understand the single land use assigned to each sampling location.
5. Line 146 – It could be good to report if any blanks tested positive for stream organisms, and if so, what was done.
6. Line 136 – A recent eDNA paper came out showing eDNA storage during field sampling occurs. It might be important to address if this is a concern if the sample protocols storage day-of at ambient temperature. Takahara et al 2020 LO Methods.

7. Filter size 22um seems appropriate; three replicate 1L with a black are appropriate methods for this work. Did the filters clog? Clogging is an issue high suspended sediment water. Some report inhibition.
8. The manuscript does not report how the three triplicates were analyzed. Pooled and averaged? It might be possible to treat them as replicates and use the GLM statistical approach.
9. What was done when a OTU was only able to be identified to a taxonomical level above species? Did the Moog data have all functional feeding groups in the genetic data? If not, what was done?
10. Line 221 – I assume community means alpha and beta diversity.
11. I think the PCA graph is important as a supplementary figure.
12. Line 228 – What variable(s) correlated with the PCA Axis 1? This could be a results; however since it is talked about here, it might be easier to just state the what landscape gradient is.
13. Line 234 – Use of Moran's I. It is unclear to me the application of Moran's I to test for spatial autocorrelation. I understand why it is used, but for rivers systems it does not make sense since network distance is probably more important than geographic distance. Plus, high spatial autocorrelation might be what the is trying to be shown. More explanation here might be warranted. The results presented on Line 244 offer the details but it difficult to understand what correlations with Moran's I here means.
14. Given the numerous statistical test being made in the analysis, performing a Bonferroni correction is warranted. Assuming a p-value of 0.05, although not stated, could lead to Type 1 errors.
15. Line 312 – The later part of this paragraph really gets into the important comparison of how the different methods might influence management decisions. The eDNA sensitive is a key finding but arguably more important is potential influence on determining ecosystem integrity assessments and follow up actions.
16. Line 341 – Interesting storyline. The material is presented leads to a localized story and potentially not appropriate for a general journal.
17. Line 395 – Needs a citation. More information to this point is necessary to make it understandable. Why are traditional methods not replaceable given the sensitivity of eDNA?
18. This paper needs a read: <https://www.nature.com/articles/s41467-020-17337-8>

Reviewers' comments:

Reviewer #1 (Remarks to the Author):

Seymour et al. used eDNA and traditional kick-net sampling to investigate how the alpha, beta and functional diversity of macroinvertebrates change across seasons and landuse types. To achieve this goal, they sampled 14 sites in 2017, with one sampling event per season (spring, summer, fall, winter) across 5 landuse types. This sampling campaign generated a total of 168 eDNA and 56 kick-net samples. The eDNA samples were amplified by targeting the COI gene and pair-end sequenced using Illumina HiSeq technology while the kick-net samples were morphologically identified to the lowest possible taxonomic level. They found that eDNA derived genera richness was significantly higher than traditional kick-net derived richness across seasons and landuse types. There was no significant effect of landuse type on both eDNA and traditional kick-net derived genera richness. They conclude that eDNA is an effective method to assess temporal and spatial patterns of macroinvertebrate diversity, and that this technique can be used to design effective biodiversity assessment and management strategies.

The idea behind the paper is great and worth investigating. The objectives of the paper, though incomplete, are clear and the hypotheses are testable. The sampling design is perfect and the number of samples collected is impressive. My main worries with this manuscript are related to the way the analyses were done and the way the results are presented. These need to be carefully addressed in order to improve the overall quality of the manuscript.

RESPONSE: Thank you for your positive assessment of our study design and sampling strategy. We have clarified the methods and results presentation to alleviate the concerns addressed below.

Specific comments

1. Lines 82-86: This statement is true but please provide relevant references to back up this claim, i.e. studies showing that eDNA is more sensitive than traditional surveys [1-5]. Please also note that this is not always true in all individual cases (reviewed in Beng & Corlett 2020 [6]). There are cases where traditional surveys outperform eDNA and where both traditional surveys and eDNA show similar performance [7-9]. In addition to the taxa- or species-specific differences in sensitivity between eDNA and traditional surveys, the environment, time of the year, and biotic factors also play important roles [10-16]. In aquatic ecosystems, for example, eDNA can persist from a few hours to a month after release [10, 11]. Differences in eDNA persistence can occur even within the same environment, for example, between the surface and bottom layers of a water body [12, 13, 16].

We have provided additional support for eDNA sensitivity (line 79-84), included the Beng and Corlett review citation as a caveat to the first point (line 79-84), and citations for the influence of environmental variation on eDNA capture (line 79-88).

2. Lines 97-99: The objective “We investigate whether metabarcoding of headwater river eDNA is effective for the assessment of macroinvertebrate community diversity dynamics across season and landuse types” is clearly stated but this was not the sole objective of the study. **Please list all objectives sequentially and perhaps match each objective with the 3 hypothesis below.**

3. Hypothesis 1: With regards to seasonal communities of riverine macroinvertebrates, richness or alpha diversity, is expected to peak during spring and summer, when many stream macroinvertebrates are emerging as adults and reproducing, compared to fall and winter months when total biomass of many species has declined (Lines 99-103).

4. Hypothesis 2: Environmental filtering effects that are linked to habitat modification, such as agriculture or urbanized areas, are expected to negatively impact macroinvertebrate diversity, particularly Chironomidae and Ephemeroptera, Plecoptera and Trichoptera (EPT) taxa, and lead to homogenized functional diversity through the combined processes of environmental filtering and biotic interactions (Lines 103-107).

5. Hypothesis 3: Utilizing the nested and turnover components of inter-community similarity (i.e. beta-diversity), we can expect high turnover within sites as community assembly changes over time, and high nestedness across environmental sites, attributed to environmental filtering of the localized sites. Alternatively, low nestedness could indicate a low effect of environmental filtering and a greater effect of stochastic or biotic factors influencing the localized community assemblies (Lines 107-112).

RESPONSE: Thank you, the text has been edited to clarify as suggested lines 116-138

6. The sampling scheme was well designed and the number of samples (168 eDNA and 56 traditional kick-net) collected is impressive. Thank you for investing so much time and effort to generate such as a comprehensive dataset.

RESPONSE: Thank you

7. Lines 191-193: Usearch has an algorithm (UCHIME) for denovo chimera detection [17]. I am curious to know if chimera detection/removal was performed in this study and if not, how were chimeric sequences detected and removed.

RESPONSE: Chimera detection and removal was performed as part of the bioinformatics pipeline. This is now clarified on line 449-451

8. Lines 193-194: Please include a reference for NCBI-BLAST [18]. I assume that default settings (e.g. Max target sequences = 100, Expected threshold = 0.05, Word size = 28, etc) were used but please mention it here, and if not please state any changes or modifications.

RESPONSE: A reference has been added (line 449-451) and the use of default settings has been specified on line 449-451

Coordinators, N. R. (2016). Database resources of the National Center for Biotechnology Information. Nucleic Acids Research, 44(D1), D7–D19. <https://doi.org/10.1093/nar/gkv1290>

9. What was the total number of OTUs obtained in this study and were OTUs present in only one sample (singletons) or in a few samples removed prior to downstream analyses? It will be interesting to also report the total number of species detected using kick-net.

RESPONSE: We have provided/clarified the total number of OTUs obtained, the treatment of singletons, and the total number of species detected using the kick net method.

10. Lines 195-200: What eventually happened to these 676 contaminant reads? It could happen that contamination in the negative controls is also present in the actual samples. In this case, all OTUs shared between negative controls and actual samples should be discarded prior to downstream analyses.

RESPONSE: The only potential contaminant of note was the single dipteran OTU, which was removed from the analysis. The other contaminants were non-target single cell or fungal entities and were removed by default. This is now clarified on line 158-164.

11. What is the difference between the sections 'Metacommunity diversity measures' (Line 202) and 'Statistics' (Line 218)? Both contain information on how 'data' was analyzed. To improve readability and reduce complexity, I would merge these as a single section and call it 'Data analysis'.

RESPONSE: Thank you, we have done this.

12. Lines 204-205: What was the 'lowest replicate level', 10,000, 20,000 or 30,000 reads? Please provide a value so that readers can have an overview of the quality and quantity of data being analyzed.

RESPONSE: We have clarified the statement, with additional bioinformatics and taxonomic summary statistics (line 157-171). In short most were above 10 000 with four sites below that were shown to be random, so we included them to preserve the balance of the experiment.

13. Lines 205-206: I am sorry but I don't understand the reason behind this sort of analysis. Why was this necessary and is the result from this analysis available somewhere? If not please provide this result in the supplementary material.

RESPONSE: We have added additional information to help clarify on lines 49-58, 86-92, 461-467. In short functional diversity allows for a more direct assessment of the link between biodiversity and the environment, particularly for macroinvertebrate feeding groups which is a long-standing research focus in ecology.

14. Lines 206-209: In most cases, species richness is used as a metric to measure alpha diversity. Here, genera richness was used. It would be nice to at least state why genera instead of species richness were used. A possible explanation could be that genera richness is a surrogate of species richness and that this has been shown for many taxa [19-23].

RESPONSE: alpha diversity or richness is not restricted to species level assessment. Most studies do not use species level information, particularly studies that assess freshwater macroinvertebrates which are often mixed classification up to family level.

15. Lines 209-215: The definition of functional richness here is too simple and the overall results are not at all different from genera richness (Fig 4). It's like doing the same analysis but calling them different names. I recommend looking at functional richness from a more robust perspective [24, 25] and using metrics that are not highly dependent on taxonomic richness. The following R packages might be useful. (i) cati (<https://cran.r-project.org/web/packages/cati/cati.pdf>), (ii) funrar (<https://cran.r-project.org/web/packages/funrar/funrar.pdf>), (iii) FD (<https://cran.r-project.org/web/packages/FD/FD.pdf>). But feel free to explore other alternatives.

RESPONSE: These are well established functional groups, particularly for linking freshwater macroinvertebrate function to ecosystem function (lines 100-108). The functional assignments are not renaming of individual, they are partitioning of functional scores within and among community members (line 461-467). The suggested alternative methods still require functional assignments to be made to community members. We have kept the analyses as it provides a direct assessment of the freshwater macroinvertebrate functionality following existing research terminology and avoids issues with interpreting transformed response variables, as with the suggested alternative methodologies.

16. Lines 215-216: The citation Baselga (2010) is missing from the list of references [26]. Stating that "Beta diversity was calculated as nestedness and turnover components following Baselga (2010)" is not enough. Please provide details of the functions used (e.g. 'beta-

multi.R', 'beta-pairwise.R' or both), state whether resampling was involved or not and how many random samples were used.

RESPONSE: The calculations were taken directly from Beselga et al (2010), with the formula now included in the text. We did not use additional R functions. We reported the full standard error for each set of pairwise distances to show the full range of the available observations, instead of using resampling to generate a mean value to allow a more direct observation of the results (Figure 5).

17. Line 221: What is 'community'? I guess this refers to 'Genera richness' as presented in the tables and figures. If yes, then please keep it consistent throughout the manuscript. Either use 'community' or 'Genera richness' throughout but not both. In each case, a clear definition from the beginning would be useful. e.g. Community refers to the total number of genera in the whole dataset. The same for 'landuse gradient' (line 227) and 'landscape gradient' (line 228). Using different terms to refer to the same thing is confusing to the readers.

RESPONSE: We have edited this section for clarity, including using richness, turnover and nestedness throughout to avoid confusion with the terms community, alpha or beta where needed.

18. Line 223: How about using 'eDNA or kicknet' instead of 'eDNA or traditional'? The former reminds readers of the specific techniques used whereas 'traditional' could mean 'anything'.

RESPONSE: We have kept the term traditional to refer to taxonomic based approaches to biodiversity assessment.

19. Line 228: 'first principal ...' or 'first principle ...'? It is not clear why the first principal component of a PCA was used in linear regression instead of using the individual non-covarying environmental variables. If I remember correctly, the samples were collected from 5 different landuse types (urban, agriculture, acid grassland, moorland, and forest), and these already represent a 'gradient'. Besides, there is no mention of how well PC1 captured this so-called 'landuse/landscape gradient' i.e. how much of the variation in the dataset was captured by PC1? Which environmental variables were highly loaded on PC1? Did PC2 also explain a considerable proportion of the variance? I would present these PCA results in the 'Results' section instead of presenting Table 1 and Figure 2. They contain valuable information that could be potentially used to interpret the results of the linear regression.

RESPONSE: We have clarified the amount of explained variation already on line 491-498. PC2 (17%) accounted for a substantially less amount of the total variation compared to PC1 (67%) and was not included in the analyses.

20. Lines 241-242: What is the importance of providing summary statistics of environmental variables (Table 1 and Figure 2) that were never used in any of your analyses and which do not directly address the objectives of this study? Since these individual variables were reduced using PCA, presenting results of the PCA makes more sense as readers can use these PCA results to better understand or interpret the linear regression results.

RESPONSE: We have provided the information as it is integral to the PCA analysis. We have expanded the PCA description (lines 500-507)

21. Lines 241-243: The presentation style of these results needs to be reconsidered. In the 'Results' section, authors are supposed to first summarize 'what they found' before referring readers to Tables and/or Figures (if any). It is strange to see statements like "All environmental variables and their associated summary statistics are presented in Table 1 &

Figure 2. Variation in genera alpha, Chironomidae and EPT richness is presented in Figure 3. The full breakdown of genera per landuse type can be found in supplementary table S1” without first summarizing the results. For instance, a summary statement such as “On average, pH was relatively lower in moorland while Gravel was relatively higher in Urban than in other landuse types (Fig 2). Or “Moorland had the lowest mean pH, Gravel and Sand, and the highest Depth, Moss and Boulder compared to the other landuse types (Fig 2). Besides, if Fig 2 must be presented, please perform an ANOVA test to compare the mean of the environmental variables across landuse types. If there’s a significant effect of landuse, please proceed to perform multiple pairwise comparisons to determine if the mean between specific pairs of landuse (e.g. Acid grassland vs Agriculture) are statistically significant (<http://www.sthda.com/english/wiki/one-way-anova-test-in-r>). You can do the same for Figure 3. Simply presenting boxplots is not enough given that readers need to know if the observed differences (genera, Chironomidae, EPT, and functional diversity) are statistically significant.

RESPONSE: We have altered the presentation style as suggested.

22. The authors presented some interesting hypotheses at the end of the Introduction (lines 99-112) but these have not been satisfactorily tested. To test these hypotheses, ANOVA and multiple pairwise comparisons are inevitable (See the link in the previous comment on how to perform such analyses).

RESPONSE: We have clarified the link between the hypotheses and the appropriate statistical test, apologies for the confusion.

23. Lines 246-247: It is simply reported that the Moran’s I test for spatial autocorrelation was non-significant or significant but no further details are provided on how the ‘significant’ Moran’s I (which signifies strong spatial autocorrelation) was dealt with. Normally, this ‘significant’ Moran’s I should be included in linear regression to account for spatial autocorrelation [27-29] but it looks like this was not the case here. I wonder why it was computed in the first place.

RESPONSE: We have removed the Moran’s I test text and incorporated a gls framework to account for any spatial autocorrelation in the data (lines 486-497).

24. Lines 249-270: The section heading “Method comparison” is misleading. Which methods are being compared here? The contents of this section are the results of linear regression and **naming it “Drivers of taxonomic and functional richness”** would be reasonable. Spatial autocorrelation should be accounted for in these analyses and if possible use the individual non-covarying environmental variables for easy interpretation. Please make a clear distinction between the ‘Landuse’ in Table 2 and in Figure 3. It’s worth emphasizing in the title that the ‘Landuse’ in Table 2 represents or was obtained from PCA.

RESPONSE: We have removed this section, while incorporating and expanding the text as part of the community dynamics section of the results. We have incorporated a more direct statistical framework to account for spatial autocorrelation, given the confusion the Moran’s I analysis created.

25. Lines 250-251: I guess there is a typo in this sentence “Genera richness derived from eDNA was significantly greater than traditional derived eDNA across season ($p < 0.01$) and landuse type ($p < 0.001$). The second ‘eDNA’ should be ‘richness’? In short, please rephrase.

RESPONSE: Thank you, this has been corrected

26. Lines 261-270: Judging from the title of the manuscript, these are the key findings. Here, it would be good to first present a figure showing the partitioning of beta diversity into

turnover and nestedness components for eDNA and kicknet together. Similar graphs for genera, Chironomidae, EPT, and functional richness would be a plus. Perhaps Figure 2 of Baselga & Orme [30] can be of help.

RESPONSE: We have included a new figure (figure 5) that shows the partitioning of nestedness and turnover

27. Line 272: These analyses could be improved further using ANOVA and multiple pairwise comparisons tests. See previous comments above.

RESPONSE: These tests have been updated and clarified with the restructuring of the results and the analyses requested by the reviewers

28. Lines 71-72: This sentence should end with a full stop (.) and not a comma (,)

RESPONSE: Thank you, this has been corrected

29. Lines 73-77: These sentences are currently difficult to read, please consider rephrasing them. "Taxonomic identification of freshwater macroinvertebrates for biomonitoring is largely limited to mature life stages that (and) can be (difficult) different among taxonomically similar species or genera. The high level of taxonomic specialization required to identify specimens and the long processing times per traditional sample (rendering) render large scale ecosystem-wide traditional assessments expensive and time consuming"

RESPONSE: We have edited this section for clarity

References

1. Kraaijeveld, K., et al., Efficient and sensitive identification and quantification of airborne pollen using next-generation DNA sequencing. *Molecular Ecology Resources*, 2015. 15(1): p. 8-16.
2. Olds, B.P., et al., Estimating species richness using environmental DNA. *Ecology and Evolution*, 2016. 6(12): p. 4214-4226.
3. Deiner, K., et al., Environmental DNA reveals that rivers are conveyor belts of biodiversity information. *Nature Communications*, 2016. 7: p. 12544.
4. Tingley, R., et al., Environmental DNA sampling as a surveillance tool for cane toad *Rhinella marina* introductions on offshore islands. *Biological Invasions*, 2019. 21(1): p. 1-6.
5. Strickland, G.J. and J.H. Roberts, Utility of eDNA and occupancy models for monitoring an endangered fish across diverse riverine habitats. *Hydrobiologia*, 2019. 826(1): p. 129-144.
6. Beng, K.C. and R.T. Corlett, Applications of environmental DNA (eDNA) in ecology and conservation: opportunities, challenges and prospects. *Biodiversity and Conservation*, 2020. 29(7): p. 2089-2121.
7. Hanfling, B., et al., Environmental DNA metabarcoding of lake fish communities reflects long-term data from established survey methods. *Molecular Ecology*, 2016. 25(13): p. 3101-3119.
8. Yamamoto, S., et al., Environmental DNA metabarcoding reveals local fish communities in a species-rich coastal sea. *Scientific Reports*, 2017. 7.
9. Hopken, M.W., et al., Molecular forensics in avian conservation: a DNA-based approach for identifying mammalian predators of ground-nesting birds and eggs. *BMC Research Notes*, 2016. 9(1): p. 14.
10. Dejean, T., et al., Persistence of Environmental DNA in Freshwater Ecosystems. *Plos One*, 2011. 6(8).
11. Pilliod, D.S., et al., Factors influencing detection of eDNA from a stream-dwelling amphibian. *Molecular Ecology Resources*, 2014. 14(1): p. 109-116.
12. Lacoursière-Roussel, A., et al., eDNA metabarcoding as a new surveillance approach for coastal Arctic biodiversity. *Ecology and evolution*, 2018. 8(16): p. 7763-7777.

13. O'Donnell, J.L., et al., Spatial distribution of environmental DNA in a nearshore marine habitat. *PeerJ*, 2017. 5.
14. Barnes, M.A. and C.R. Turner, The ecology of environmental DNA and implications for conservation genetics. *Conservation Genetics*, 2016. 17(1): p. 1-17.
15. Takeuchi, A., et al., Release of eDNA by different life history stages and during spawning activities of laboratory-reared Japanese eels for interpretation of oceanic survey data. *Scientific Reports*, 2019. 9(1): p. 6074.
16. Anglès d'Auriac, M.B., et al., Detection of an invasive aquatic plant in natural water bodies using environmental DNA. *PLOS ONE*, 2019. 14(7): p. e0219700.
17. Edgar, R.C., et al., UCHIME improves sensitivity and speed of chimera detection. *Bioinformatics (Oxford, England)*, 2011. 27(16): p. 2194-2200.
18. Altschul, S.F., et al., Basic local alignment search tool. *Journal of Molecular Biology*, 1990. 215(3): p. 403-410.
19. Andersen, A.N., Measuring more of biodiversity: Genus richness as a surrogate for species richness in Australian ant faunas. *Biological Conservation*, 1995. 73(1): p. 39-43.
20. Balmford, A., A.H.M. Jayasuriya, and M.J.B. Green, Using Higher-Taxon Richness as a Surrogate for Species Richness: II. Local Applications. *Proceedings: Biological Sciences*, 1996. 263(1376): p. 1571-1575.
21. Báldi, A., Using higher taxa as surrogates of species richness: a study based on 3700 Coleoptera, Diptera, and Acari species in Central-Hungarian reserves. *Basic and Applied Ecology*, 2003. 4(6): p. 589-593.
22. Alves, C., et al., Genera as surrogates of bryophyte species richness and composition. *Ecological Indicators*, 2016. 63: p. 82-88.
23. Balmford, A., M.J.B. Green, and M.G. Murray, Using higher-taxon richness as a surrogate for species richness: I. Regional tests. *Proceedings of the Royal Society of London. Series B: Biological Sciences*, 1996. 263(1375): p. 1267-1274.
24. Laureto, L.M.O., M.V. Cianciaruso, and D.S.M. Samia, Functional diversity: an overview of its history and applicability. *Natureza & Conservação*, 2015. 13(2): p. 112-116.
25. Hatfield, J.H., M.L.K. Harrison, and C. Banks-Leite, Functional Diversity Metrics: How They Are Affected by Landscape Change and How They Represent Ecosystem Functioning in the Tropics. *Current Landscape Ecology Reports*, 2018. 3(2): p. 35-42.
26. Baselga, A., Partitioning the turnover and nestedness components of beta diversity. *Global Ecology and Biogeography*, 2010. 19(1): p. 134-143.
27. Bispo, P.d.C., et al., Drivers of metacommunity structure diverge for common and rare Amazonian tree species. *PLOS ONE*, 2017. 12(11): p. e0188300.
28. Liu, J.-J. and J.W.F. Slik, Forest fragment spatial distribution matters for tropical tree conservation. *Biological Conservation*, 2014. 171: p. 99-106.
29. Slik, J.W.F., et al., Large trees drive forest aboveground biomass variation in moist lowland forests across the tropics. *Global Ecology and Biogeography*, 2013. 22(12): p. 1261-1271.
30. Baselga, A. and C.D.L. Orme, betapart: an R package for the study of beta diversity. *Methods in Ecology and Evolution*, 2012. 3(5): p. 808-812.

Reviewer #2 (Remarks to the Author):

Review of Seymour et al – Nestedness and turnover of riverine species and functional diversity using eDNA and traditional approaches.

This manuscript describes the use of eDNA and traditional sampling methods to assess macroinvertebrate community diversity. In addition the effects of land use and seasonality on community and functional diversity is assessed. This is a very thorough study with interesting findings which will strengthen knowledge in this area and could be used to inform management strategies.

I think that the statistical analysis presented is of sound quality and appropriate to this study. A small amount of additional detail in the methods section (see below) will further make this study reproducible.

I have a number of comments to make many of which are minor:

1. In the title I don't think you can use 'species' as you are discussing genus/order/family level changes. You could use 'macroinvertebrates'. I think you may also need to be clear as to the level to which you are referring in the abstract as initially I thought you would be discussing results down to the species level.

RESPONSE: We have edited the title to remove the term species

2. Full stop needed at end of line 72.

RESPONSE: Edited, thank you

3. Line 76 'rendering' should this be 'renders'?

RESPONSE: Edited

4. Lines 82-86, very long sentence, is it possible to split into two or three sentences.

RESPONSE: We have edited this section for grammar

5. Line 88 – reference examples of this.

RESPONSE: We have edited this sentence. The intent was to link with the subsequent paragraph, which has the key references.

6. Line 93 – again any references for this statement?

RESPONSE: Yes, we have added

7. Is an ethics statement required for destructive sampling?

RESPONSE: A kicknet sample is not destructive and does not require an ethics statement.

8. Figure 1 Legend – delete 'Colors)', 'landuse' is two separate words (throughout document), and I'd add the number of sampling locations.

RESPONSE: Landuse is grammatically correct as a single word and reduces confusion from splitting it.

9. Line 131 - I assume eDNA was collected first at each site, specify this.

RESPONSE: clarified on line 389-390

10. Line 132 – was there any variation in the amount of water sampled i.e. did 1L always go through the filter okay?

RESPONSE: water samples were mostly 1L with some restricted to 500ml due to sedimentation. Clarified on line 382-387. This was previously found to not have an effect on the results from previous experiments with this methodology.

11. Line 134 – how long were bottles soaked in 10% bleach?

RESPONSE: One hour, now clarified on line 379

12. Line 136 – how much ATL buffer was used?

RESPONSE: 500 ul, clarified on line 385-387

13. Line 138 – reference for standardised protocol?

RESPONSE: we have provided the reference at line 394

***River Habitat Survey in Britain and Ireland: Field Survey Guidance Manual: 2003 Version.*
(Forest Research, 2003)**

14. Line 141-143 – how, microscopy, by eye, were any identification keys used (if so reference these).

RESPONSE: The method differs by taxa. Generally a stereoscope is sufficient, but some groups require different levels due to variation in body size.

***River Habitat Survey in Britain and Ireland: Field Survey Guidance Manual: 2003 Version.*
(Forest Research, 2003)**

15. Line 143-146 – when taken before/after eDNA? I'm asking to ensure that there was as little disturbance of the site prior to eDNA sampling as possible.

RESPONSE: eDNA samples were taken first, then kicknets, now clarified on line (389-390)

16. Line 146-150 – this is not entirely clear as you have not specified how many sampling events there were in total and you only say three field blanks yet there are 12 blanks in total, if three field blanks sampled in triplicate or three per sampling event, this would only be 9 samples. What volume are the field blanks? I think a little bit of clarification is required here. Were there any laboratory blanks or is this what you are referring to from 148 on?

RESPONSE: We had four sampling days (one per each season), with each sampling event including 14 sites. We processed three blanks for each sampling day with one blank being run every 4th site (4 X 3 = 12 blanks in total). The blanks were treated the same as the samples, so we used 1L blanks. This is clarified on line 399-401

17. Line 152 – was a unidirectional lab flow followed, you should specify this.

RESPONSE: Yes, clarified on line 405-406

18. Line 155 – volume of PK and temperature?

RESPONSE: 70 and 58 per protocol. Now clarified on line 407-409

19. Line 158 – what was the final volume/concentration of the extracted DNA, I would add this information?

RESPONSE: added at line 411

20. Line 167 – name of indexing kit used?

RESPONSE: Clarified that these are Illumina matching indices on line 421-423

21. Line 168 – remove capital ‘D’ from ‘Due’.

RESPONSE: Corrected

22. Line 176 – were the PCR1 amplicons purified prior to PCR2? Picky, I know but what make and model of PCR machines were used, I think that all of this needs to be included?

RESPONSE: Yes, clarified on line 418-423

23. Line 178 – used Qubit twice

RESPONSE: Corrected

24. Line 193 – was assignment to species or family level?

RESPONSE: To the genus level, this is now clarified on line 449-451

25. Lines 195-200 – these are results so should not be in the methods section. I also think that the results section would benefit from some data output information in tabular form although this might be better in the supporting information.

RESPONSE: We have moved this paragraph to the results section.

26. Line 213 – I think you should specify which families/genus are in each of these groups.

RESPONSE: We have provided a table (Table 2) to show the breakdown of the major groups per taxa

27. Line 215-216 – might be useful to put in a brief description of beta diversity calculation (or in supplemental information).

RESPONSE: A more detailed description and specific formula used are provided on lines (472-483)

28. Figure 3 – I think that the right side panels need to be the same size as the left side panels. Also need to add in what the dots refer to presumably outliers and if so an explanation as to why excluded.

RESPONSE: We have edited the figure and included that the dots refer to points outside the 1*5 inter-quartile range (Figure 4).

RESPONSE:

29. Table S1 – I think it would be useful to include percentage totals per family/site and method and also include unassigned percentages too. I know you have split into two separate tables and this looks good, but I can't help but think it would be easier to compare the two methods if the two methods were in one table. This is obviously a personal preference, but I was interested to see how different the results were between the two methods and there is no easy way to do this than to scroll back and forth. I do think that although not the focus of the paper that you should include more discussion on how comparable the two methods were e.g. any gaps in the eDNA data and/or representative venn diagrams (could add in around lines (302-304 or 316).

RESPONSE: We have edited Table S1 and have included a Venn diagram as figure 3 with representative statistics included as part of the results/discussion at lines (175-183)

30. Figure 4 – for panels A and B, what do the different colours represent – needs a key of the different genera so that the text lines 273-294 has some context. Where is the data for the previous season as referred to in the figure legend? I think this figure also needs more explanation, I understand that functional groups can show losses and gains as some families within e.g. scrapers will be lost and some gained compared to the previous season, for the genus graphs this is the case too but I think you need to spell this out by giving an example e.g. within the genus shown in green X and Y showed gains and Z showed a loss.

RESPONSE: We have added additional information for the legend, including adding the identification for each unique color as a legend to Figure 6

31. Lines 273-294 – I would highlight any obvious differences between the eDNA and traditional data within the results section too.

RESPONSE: These have been added at line 175-183 with the requested Venn diagram (Figure 3) referenced below as well as a table (Table 2) to show the number of genera per taxa group between sampling methods.

32. Line 307 'timepoints' is two words.

RESPONSE: Corrected

33. Line 316-319 – I would add in reference to the results figures again in the discussion to make it easy for the reader to cross check.

RESPONSE: Figure references are now incorporated

34. In line 331 you suggest why there may be greater macroinvertebrate diversity at moorland sites which sounds reasonable to me but you also explain why there is lower diversity with traditional methods in moorland sites (lines 323-326), I find this contradictory so this may need to be edited to make clear.

RESPONSE: Apologies for the confusion the second statement has been edited.

35. Line 345 – I would use 'Secondly' rather than 'And second' and also 'Firstly' in line above. Sentence is also very long and doesn't totally make sense, is there one too many 'and's i.e. 'moorland sites, the' rather than 'moorland sites and the'?

RESPONSE: Corrected

36. Lines 386-391 – I would start a new sentence at the semi-colon in line 388.

RESPONSE: Corrected

37. Line 393-394 – how does this compare to other studies that have compared traditional and metabarcoding approaches even if on different environments/taxa? I think this requires some discussion.

RESPONSE: We have included additional discussion for the comparison of traditional and eDNA based biodiversity at line (248-254,278-288)

--Signed Helen C. Rees

Reviewer #3 (Remarks to the Author):

The paper entitled “Nestedness and turnover of riverine species and functional diversity using eDNA and traditional approaches” by Seymour et al. presents a robust dataset comparing traditional field sample against eDNA methods for macroinvertebrates. The manuscript presents a detailed description of the ecological findings and meaning for ecological monitoring.

Comments

1. Comparing methods. It is not surprising to find that eDNA methods provide much higher diversity values. eDNA methods are known to be much more sensitive, or at least not that surprising. This point is important to show, particularly for benthic macroinvertebrates which are used for water quality monitoring. One large concern here is that any analysis comparing the methods is a result of this sensitivity difference and potentially the fact that eDNA samples (although 1L) represent a much large area of sampling then the field methods. **The sampled water is an integration of water upstream.** For the monitoring story line, **it seems more important to also communicate any genera that were caught in the field sample but not in the eDNA method.** Also, it seems important to mention the scale of inference between the two methods (see inference section in Goldberg et al 2016). The introduction does not overly state this point.

RESPONSE: These are important considerations that were taken into account for the study design and sampling prior to the experiment execution. We utilized headwater stream sites, which limits the effect of upstream signals diluting the localized test as any upstream transport will be from environmentally similar habitat. This is also relevant for any landuse/environmental variation we want to make, this has been clarified (lines 94-98). We have edited the introduction (lines 41-68, 70-92) and discussion (lines 230-242) to emphasize the ecological assessment of the data generated between the two methods. We included a supplementary table (Table S1) showing which genera were detected for each methodology (clarified on line 175-183).

The Results section on this does seem overly detailed and statistical, rather than really comparing how the methods comparison influences monitoring – method replacement or combined approach. A deeper discussion into the meaning of comparing the methods, might change the Discussion paragraph starting on Line 312 and 341. To be clear, I think it is debatable that there is utility in comparing data from these methods. However, for this dataset and manuscript, I think this point should be made clear, so the reader understands the issue before dropping into the results and discussion. The field data could be presented as validation not comparison.

RESPONSE: We have clarified/edited the point on lines 244-276, and via a figure (Figure 3). As the focus of the manuscript is on ecosystem assessment, the method comparisons are not the primary focus, but rather a means to generate comparable datasets to determine if the ecological inferences/analyses from the two sampling methods conclude similar (or dissimilar findings).

2. Unclear storyline. The ecological story of seasonal diversity change is interesting and confirming presumed patterns with better data. The monitoring story is a clear and very important one for the applied implications. However, I think the manuscript is missing a clear ‘game changer’ story line. See this recent paper for an example of a very similar topic: <https://www.nature.com/articles/s41467-020-17337-8> I think three are two about two

separate topics (monitoring and diversity). The conclusions are clear in the manuscript but not perhaps presented for a general audience. For a general audience, not a community ecology one, the manuscript's conclusion feels overly contextual and disciplinary. The monitoring story, if developed, could lead to significant attention (just my opinion). Given the dataset, I think a lot can be learned from this work. I look forward to more papers.

RESPONSE: We have added/clarified a discussion paragraph to provide a more general take home message regarding our findings (lines 338-356).

Smaller Points:

1. Line 312 – the point above that the methods might not be appropriate to compare.

RESPONSE: We appreciate this comment. The aim was not to directly compare specific outputs but to relate ecological and ecosystem level information from both methods. This is not clarified in the introduction and discussion.

2. Two storylines – I am struggling a bit with understanding the main message for the paper. There is a lot here and this dataset is impressive. The two storylines to me are: using of eDNA compared to traditional methods for BMI; (2) biodiversity (all of them) in stream BMI. The two are of course intertwined. I feel the manuscript could be improved by a better explanation **why diversity is important to understand in environmental monitoring**. Potentially, there are two papers here one on environmental monitoring using eDNA and one on the basic ecology.

RESPONSE: Thank you for the suggestion. We have refocused the introduction and discussion to emphasize the importance of eDNA based biodiversity assessment for monitoring and ecological assessment.

3. Line 50 – Implies that eDNA methods might be more cost effective. I do not think this is true yet, nor have I seen data on it other than our own. This is a larger issue that probably needs to be danced around a bit or better explained.

RESPONSE: From experience we respectfully disagree. Simply from this study, we were able to generate twice the biodiversity and process three times the samples at a faster speed compared to our taxonomic based assessment. Depending on the outsource cost of any given taxonomist the sequencing cost will become increasingly more cost effective the more samples that are being pooled, particularly with the continued drop in cost of sequencing and bioinformatics services that are required to generate these data.

4. Figure 1. Site were classed by the different land uses, yet it is unclear from Figure and the text how the land use classification was made for mixed land use sampling sites. This is important for riverine biodiversity locally since it is driven upstream source characteristics. Figure 1 could be improved by showing the sub-watershed boundaries and streams. Adding the land use classed for each site would also help the reader better understand the single land use assigned to each sampling location.

RESPONSE: These sites are upstream sites, selected to avoid upstream mixing of other landuse types.

5. Line 146 – It could be good to report if any blanks tested positive for stream organisms, and if so, what was done.

RESPONSE: This is reported on lines 157-171. The only curious find was a potential dipteran ASV at low contamination levels, which were removed in downstream analyses. Other blank sequences were associated with non-target bacteria, algae or fungi.

6. Line 136 – A recent eDNA paper came out showing eDNA storage during field sampling occurs. It might be important to address if this is a concern if the sample protocols storage day-of at ambient temperature. Takahara et al 2020 LO Methods.

RESPONSE: Samples were collected, preserved in the field and transported same day. We have edited the text as samples were kept in a cooler directly after sampling and were not kept at ambient temperature at any point in the process. Apologies for the confusion.

7. Filter size 22um seems appropriate; three replicate 1L with a black are appropriate methods for this work. Did the filters clog? Clogging is an issue high suspended sediment water. Some report inhibition.

RESPONSE: We clarified that all extractions were cleaned using Qiagen inhibition clean up kits. We also clarified that clogging would occur, but that we were able to get >500 ml per sample

8. The manuscript does not report how the three triplicates were analyzed. Pooled and averaged? It might be possible to treat them as replicates and use the GLM statistical approach.

RESPONSE: We have clarified this on line 455-459 of the methods. The triplicates were used to more robustly determine presence of a given genera, meaning the triplicate data were averaged to generate a site specific data point. This allowed for a balanced design with the taxonomic data for any downstream statistical analyses to keep the model assumptions.

9. What was done when a OTU was only able to be identified to a taxonomical level above species? Did the Moog data have all functional feeding groups in the genetic data? If not, what was done?

RESPONSE: Those unidentified to the genera or to a functional group were not included in the analyses

10. Line 221 – I assume community means alpha and beta diversity.

RESPONSE: Community refers to the site data and is now clarified. We have replaced alpha and beta with richness and turnover/nestedness throughout to avoid confusion.

11. I think the PCA graph is important as a supplementary figure.

RESPONSE: OK we will keep it in the supplement

12. Line 228 – What variable(s) correlated with the PCA Axis 1? This could be a results; however since it is talked about here, it might be easier to just state the what landscape gradient is.

RESPONSE: This was provided on lines 500-507.

13. Line 234 – Use of Moran's I. It is unclear to me the application of Moran's I to test for spatial autocorrelation. I understand why it is used, but for rivers systems it does not make sense since network distance is probably more important than geographic distance. Plus, high spatial autocorrelation might be what the is trying to be shown. More explanation here

might be warranted. The results presented on Line 244 offer the details but it difficult to understand what correlations with Moran's I here means.

RESPONSE: These sites are not connected via the river network as they are all head waters scattered across the river drainage with no network structure linking any two sites (no downstream movement from one site connects with another). Many macroinvertebrate species, particularly EPT and chironomid species have flying adult stages, which makes the linear distance assumption more valid as well.

The Moran's I test is a test for spatial autocorrelation, which is often overlooked in traditional studies. We have removed this section and incorporated a variance structure for spatial autocorrelation as part of the statistical models to hopefully avoid confusion.

14. Given the numerous statistical test being made in the analysis, performing a Bonferroni correction is warranted. Assuming a p-value of 0.05, although not stated, could lead to Type 1 errors.

RESPONSE: We do not perform more than one test on each subset of the data, as we specifically designed models to incorporate season, landuse and method type. Each test is independent and includes the full set of data per test. A bonferoni would be warranted if we were to conduct multiple tests of the same explanatory variable on subsets of the data, such as separating test by sampling methodologies or season.

15. Line 312 – The later part of this paragraph really gets into the important comparison of how the different methods might influence management decisions. The eDNA sensitive is a key finding but arguably more important is potential influence on determining ecosystem integrity assessments and follow up actions.

RESPONSE: We have substantially edited the discussion and provide more direct points to highlight how these data are important for understanding ecosystem status and management approaches (lines 330-336)

16. Line 341 – Interesting storyline. The material is presented leads to a localized story and potentially not appropriate for a general journal.

RESPONSE: We have clarified this to include a broader context.

17. Line 395 – Needs a citation. More information to this point is necessary to make it understandable. Why are traditional methods not replaceable given the sensitivity of eDNA?

RESPONSE: We have expanded this to provide additional clarification and have removed this sentence to avoid confusion with our main results and findings.

18. This paper needs a read: <https://www.nature.com/articles/s41467-020-17337-8>

RESPONSE: Again, we appreciate the Carraro piece (transport, network structure, edna only), however the study design and question are not related to our study (headwaters, landuse, taxonomic identification). We have cited it none-the-less.

REVIEWERS' COMMENTS:

Reviewer #1 (Remarks to the Author):

General comments

I would like to thank Seymour et al for their effort to address the issues raised in the previous version of this manuscript. The current version has substantially improved. However, I would like to invite the authors to devote some attention to typos that might hinder or distract readers from understanding the main message of the manuscript. I also don't understand why the authors insist on referring to their study group as "biodiversity" instead of "macroinvertebrates". Some statements sound as if the authors quantified all living organisms in their study but they actually only looked at macroinvertebrates. Why not call a spade a spade? Similarly, the authors insist on referring to "kick-net sampling" as "traditional methods" whereas they could just refer to it as such or as traditional kick-net. Their use of traditional methods/approaches all throughout the manuscript sounds like they applied more than one method/approach. It is difficult for the reader to focus in, as one always need to stop, puzzle for a while and say OK the paper is about macroinvertebrates, eDNA, and kick-netting. If I single word or few phrases can be used to send a message across, please avoid using multiple or complex phrases.

Specific comments

Line 2: Only one traditional approach (kick-net sampling) was used, right? Should this be "traditional approaches"?

Lines 14-16: If the abstract acts as a stand-alone summary of the manuscript, readers expect to know which specific "community" and "traditional methods" were used, without reading the whole manuscript. At least tell the readers that you used headwater or freshwater macroinvertebrates and kick-net sampling. Did you really "assess spatio-temporal dynamics between eDNA and traditional methods"? How can one assess the spatio-temporal dynamics of "methods"? Shouldn't it be the other way round that you used eDNA and kick-net sampling to assess the spatio-temporal dynamics (community richness and functional diversity) of headwater or freshwater macroinvertebrates? You prefer the term "traditional method", yes, but this is just one out of N traditional methods. Again, one traditional method was applied; please take note to use the singular (traditional method) throughout the manuscript.

Lines 16-17: What does "greater biodiversity resolution" mean? A is greater than B! What are these complementary findings? Again, please remember that the abstract needs to be stand-alone.

Line 23: If you are referring to the advantage of eDNA, how about using "... previously not possible with traditional methods instead of "... previously unavailable by traditional means"?

Line 32-33: Are there non-living biological communities? I know of "living organisms" or just using "biological communities" should be enough. In fact, I don't get the point the entire sentence is trying to make. "... change in living biological communities (e.g. biodiversity) to assess changes in ecosystem...?"

Line 39: promote or improve ecosystem function and health?

Line 80: eDNA capture can differ between "richness"? Please clarify

Lines 84-86: If a single word can communicate the message, please avoid using multiple words/phrases, e.g. "eDNA-based studies" and "eDNA based sampling" could be simply expressed as "eDNA". Depending on the context, it should be clear that eDNA refers to studies and/or sampling approaches. Sometimes you use "eDNA-based" (with hyphen) and other times "eDNA based" (without hyphen). Please be consistent. The same use of multiple words/phrases applies to "ecological monitoring practices", which could simply be written as "ecological monitoring" without

changing the meaning of the statement. There are other instances throughout the manuscript, please check and simplify.

Lines 98-100: "...source of ecosystem assessment information..." sounds confusing. Please consider rephrasing, "Freshwater macroinvertebrates are an invaluable source of information for ecosystem assessment..."

Line 106: feeding groups or feedings groups?

Line 109: Is "and" necessary in this sentence "however, is largely limited to mature life stages that and can be difficult to identify or"? I suggested checking this in the previous version but all 10 authors missed it? "a" is also NOT necessary in line 113.

Line 122: eDNA by itself is NOT biodiversity, it's a technique. What you mean in one is that eDNA will capture higher macroinvertebrate biodiversity than kick-net sampling? The study was limited to macroinvertebrates; using "biodiversity" in its broad sense does not help the readers a lot.

Line 123: Using "riverine macroinvertebrate biodiversity" in the first objective/hypothesis and "localized community richness" in the second will be much easier to follow.

Line 129: "environmental sites" could just be "sites" and "environmental filtering of the localized sites" could simply read "environmental filtering" and convey the same message.

Lines 140-151: Are these results or did I miss something? Reporting results in the introduction

Lines 176-177: Finally, "eDNA and traditional kick-netting" are used explicitly, instead of "eDNA and traditional methods" used in the preceding sections. Great job!

Line 185: What is traditional derived eDNA?

Lines 187-188: This is NOT biodiversity dynamics, this is macroinvertebrate dynamics!

Line 199: is "or" necessary here? Did you mean to say eDNA or traditional methods?

Line 201: double "greater", please delete one.

Line 202: "than traditional methods" or "versus traditional methods"?

Line 231: spatial and temporal dynamics of what? Freshwater macroinvertebrates

Line 269: it's "...upstream transport limited..." NOT "...upstream transported limited..." please

Line 296: it's "...very strong..." NOT "...very stronger..." please

Line 394: it's "...lowest practical taxonomy..." NOT "...lowest practical taxonomic..." please

Line 491: it's "...by including..." NOT "...by included..." please

Reviewer #2 (Remarks to the Author):

I think that this paper has been much improved by the various additions to the manuscript. I have specifically looked at the responses to my original comments and not that of the other reviewers and am happy with the changes that have been made. My only very minor comments are that there are still quite a few very long sentences in the manuscript which could benefit from a general edit e.g. in the objectives and hypotheses section.

In line 201 there are two 'greater's.

In line 253 'though see Leese....' suggests that you are going to go on and say something about that paper but you haven't?

Reviewer #2 (Remarks to the Author):

Please see attachment

Review of Seymour et al. Manuscript Round 2

Thank you for the opportunity for reviewing the revised manuscript by Seymour et al. now entitled “Environmental DNA provides greater insight to biodiversity and ecosystem function compared to traditional approaches, via spatio-temporal nestedness and turnover partitioning”. The motivation, main hypothesis, findings, and conclusions are clearer in the revised manuscript, particularly the revisions to the Introduction. I return to a point I made previously about ‘cost effectiveness’ below. I feel it an important consideration for this paper and needs, at least, clarification.

Alexander Fremier

Response to rebuttal comments.

*2. Two storylines – I am struggling a bit with understanding the main message for the paper. There is a lot here and this dataset is impressive. The two storylines to me are: using of eDNA compared to traditional methods for BMI; (2) biodiversity (all of them) in stream BMI. The two are of course intertwined. I feel the manuscript could be improved by a better explanation **why diversity is important to understand in environmental monitoring**. Potentially, there are two papers here one on environmental monitoring using eDNA and one on the basic ecology.*

RESPONSE: Thank you for the suggestion. We have refocused the introduction and discussion to emphasize the importance of eDNA based biodiversity assessment for monitoring and ecological assessment.

I appreciate the time and thought that went into the re-write of the introduction. The revisions satisfy my multiple comments about the refocusing the manuscript on genetic based biodiversity techniques in monitoring, rather than a comparison paper. The transparent hypotheses R1 requested further support the revised manuscript focus. The figure adjustment also clarifies the data supporting discussion points. The first paragraph in the Discussion returns to the Main Points with clear and objective findings from the data. I think this paragraph will help the reader pull out the main findings. I think the revised manuscript has a clearer focus and structure.

3. Line 50 – Implies that eDNA methods might be more cost effective. I do not think this is true yet, nor have I seen data on it other than our own. This is a larger issue that probably needs to be danced around a bit or better explained.

RESPONSE: From experience we respectfully disagree. Simply from this study, we were able to generate twice the biodiversity and process three times the samples at a faster speed compared to our taxonomic based assessment. Depending on the outsource cost of any

given taxonomist the sequencing cost will become increasingly more cost effective the more samples that are being pooled, particularly with the continued drop in cost of sequencing and bioinformatics services that are required to generate these data.

I was probably not as clear in my initial review as I should have been. My comments on the cost efficiency/effectiveness is not about the amount of data produce by either method. Clearly, genetic based methods produce more data. The point here in the manuscript is about monetary cost. I have not read a paper that has compared genetic based versus field-based methods for cost efficiency. That is, for the same amount of data (e.g., benthic macroinvertebrate abundance data of indicator species only) which method is better? The authors argue, as I think I would as well, that genetic methods are more cost efficient than field methods. My point here is that a citation would be necessary to confirm this, or data from this work to show it. The later seems out of the scope of this paper. If there are no citations available, then perhaps the claim of cost efficiency should be clarified. This is obviously not a deal breaker for publication, but it is an important assumption that many pass over which has big implications for potential replacement monitoring techniques.

Line 13 – “while reducing cost and time”. I want to believe this but I honestly do not think this direct claim is precise. Is there a cost comparison research paper out there yet? Not be my knowledge. Certainly, eDNA produces more data and data that field methods cannot produce. But from a strict monitoring needs driven by policy, field methods might produce the data the policy requires (e.g., high EPT taxa in a sample) at a low cost. Plus, eDNA has not been shown to reliably estimate population size. qPCR estimates are a start, but they cannot replace areal based estimates of field surveys. I do not see eDNA a replacement method yet, but it certainly has the potential. It is probably best described as a complementary method currently. The latter half of the Abstract is on point – it details the benefits and does not wade into the ‘cost’ complications that this paper does not directly address.

Line 140 – “better descriptor” perhaps change to “more complete”

Line 268 – “With upstream transported limited above” - typo? Overall, this sentence is difficult to understand and a very important point.

Line 330 to the end – Difficult recommendations to follow.

Line 335 head water to headwater

Line 338 to the end – Clear conclusion linking back to the Introduction.

REVIEWERS' COMMENTS:

Reviewer #1 (Remarks to the Author):

General comments

I would like to thank Seymour et al for their effort to address the issues raised in the previous version of this manuscript. The current version has substantially improved. However, I would like to invite the authors to devote some attention to typos that might hinder or distract readers from understanding the main message of the manuscript. I also don't understand why the authors insist on referring to their study group as "biodiversity" instead of "macroinvertebrates". Some statements sound as if the authors quantified all living organisms in their study but they actually only looked at macroinvertebrates. Why not call a spade a spade? Similarly, the authors insist on referring to "kick-net sampling" as "traditional methods" whereas they could just refer to it as such or as traditional kick-net. Their use of traditional methods/approaches all throughout the manuscript sounds like they applied more than one method/approach. It is difficult for the reader to focus in, as one always need to stop, puzzle for a while and say OK the paper is about macroinvertebrates, eDNA, and kick-netting. If I single word or few phrases can be used to send a message across, please avoid using multiple or complex phrases.

RESPONSE: Thank you for your review. We have checked the manuscript to ensure terminology is consistent throughout the manuscript.

Specific comments

Line 2: Only one traditional approach (kick-net sampling) was used, right? Should this be "traditional approaches"?

RESPONSE: The title no longer includes the term.

Lines 14-16: If the abstract acts as a stand-alone summary of the manuscript, readers expect to know which specific "community" and "traditional methods" were used, without reading the whole manuscript. At least tell the readers that you used headwater or freshwater macroinvertebrates and kick-net sampling. Did you really "assess spatio-temporal dynamics between eDNA and traditional methods"? How can one assess the spatio-temporal dynamics of "methods"? Shouldn't it be the other way round that you used eDNA and kick-net sampling to assess the spatio-temporal dynamics (community richness and functional diversity) of headwater or freshwater macroinvertebrates? You prefer the term "traditional method", yes, but this is just one out of N traditional methods. Again, one traditional method was applied; please take note to use the singular (traditional method) throughout the manuscript.

RESPONSE: We have edited the text to indicate one traditional method was used.

Lines 16-17: What does "greater biodiversity resolution" mean? A is greater than B! What are these complementary findings? Again, please remember that the abstract needs to be stand-alone.

RESPONSE: We have edited the text to clarify, and now reads, "Environmental DNA (eDNA) based assessment offers enhanced scope for assessing biodiversity, while increasing sampling efficiency and reducing processing time, compared to traditional methods."

Line 23: If you are referring to the advantage of eDNA, how about using "... previously not possible with traditional methods instead of "... previously unavailable by traditional means"?"

RESPONSE: Thank you we have edited the sentence as suggested

Line 32-33: Are there non-living biological communities? I know of "living organisms" or just using "biological communities" should be enough. In fact, I don't get the point the entire sentence is trying to make. "... change in living biological communities (e.g. biodiversity) to assess changes in ecosystem...?"

RESPONSE: We have edited the sentence to avoid repetitiveness.

Line 39: promote or improve ecosystem function and health?

RESPONSE: As improve is a synonym of promote we have kept the text as is.

Line 80: eDNA capture can differ between "richness"? Please clarify

RESPONSE: This sentence has been removed to avoid confusion.

Lines 84-86: If a single word can communicate the message, please avoid using multiple words/phrases, e.g. "eDNA-based studies" and "eDNA based sampling" could be simply expressed as "eDNA". Depending on the context, it should be clear that eDNA refers to studies and/or sampling approaches. Sometimes you use "eDNA-based" (with hyphen) and other times "eDNA based" (without hyphen). Please be consistent. The same use of multiple words/phrases applies to "ecological monitoring practices", which could simply be written as "ecological monitoring" without changing the meaning of the statement. There are other instances throughout the manuscript, please check and simplify.

RESPONSE: We have removed the hyphenated "eDNA-based" term throughout and simply use "eDNA."

Lines 98-100: "...source of ecosystem assessment information..." sounds confusing. Please consider rephrasing, "Freshwater macroinvertebrates are an invaluable source of information for ecosystem assessment..."

RESPONSE: Thank you, we have edited the sentence as suggested

Line 106: feeding groups or feedings groups?

RESPONSE: Thank you, now edited.

Line 109: Is "and" necessary in this sentence "however, is largely limited to mature life stages that and can be difficult to identify or"?

RESPONSE: Thank you, now edited without the "and"

I suggested checking this in the previous version but all 10 authors missed it? "a" is also NOT necessary in line 113.

RESPONSE: Thank you again, the “a” is now removed.

Line 122: eDNA by itself is NOT biodiversity, it’s a technique. What you mean in one is that eDNA will capture higher macroinvertebrate biodiversity than kick-net sampling? The study was limited to macroinvertebrates; using “biodiversity” in its broad sense does not help the readers a lot.

RESPONSE: In line with your previous comment regarding eDNA terminology, we have kept the text as is since it is implied at this stage that the biodiversity is derived from the eDNA sampling.

Line 123: Using “riverine macroinvertebrate biodiversity” in the first objective/hypothesis and “localized community richness” in the second will be much easier to follow.

RESPONSE: We have changed the order as suggested

Line 129: “environmental sites” could just be “sites” and “environmental filtering of the localized sites” could simply read “environmental filtering” and convey the same message.

RESPONSE: Thank you, edited as suggested

Lines 140-151: Are these results or did I miss something? Reporting results in the introduction

RESPONSE: This paragraph is to conform with Communication Biology’s journal style requirements.

Lines 176-177: Finally, “eDNA and traditional kick-netting” are used explicitly, instead of “eDNA and traditional methods” used in the preceding sections. Great job!

RESPONSE: Thank you.

Line 185: What is traditional derived eDNA?

RESPONSE: Edited, apologies

Lines 187-188: This is NOT biodiversity dynamics, this is macroinvertebrate dynamics!

RESPONSE: Macroinvertebrates are multi-species which constitutes a level of biodiversity.

Line 199: is “or” necessary here? Did you mean to say eDNA or traditional methods?

RESPONSE: Edited, thank you

Line 201: double “greater”, please delete one.

RESPONSE: Edited, thank you

Line 202: “than traditional methods” or “versus traditional methods”?

RESPONSE: Replaced with “compared to” to avoid confusion

Line 231: spatial and temporal dynamics of what? Freshwater macroinvertebrates

RESPONSE: We have added biodiversity to clarify.

Line 269: it’s “...upstream transport limited...” NOT “...upstream transported limited...” please

RESPONSE: Edited

Line 296: it’s “...very strong...” NOT “...very stronger...” please

RESPONSE: Edited

Line 394: it’s “...lowest practical taxonomy...” NOT “...lowest practical taxonomic...” please

RESPONSE: Edited, thank you

Line 491: it’s “...by including...” NOT “...by included...” please

RESPONSE: Edited

Reviewer #2 (Remarks to the Author):

I think that this paper has been much improved by the various additions to the manuscript. I have specifically looked at the responses to my original comments and not that of the other reviewers and am happy with the changes that have been made. My only very minor comments are that there are still quite a few very long sentences in the manuscript which could benefit from a general edit e.g. in the objectives and hypotheses section.

RESPONSE: Thank you for your time and positive assessment of our work

In line 201 there are two 'greater's.

RESPONSE: Edited, thank you.

In line 253 'though see Leese....' suggests that you are going to go on and say something about that paper but you haven't?

RESPONSE: We have clarified that we meant but see Leese et al. for an exception to the fore mentioned statement.

Reviewer #3 (Remarks to the Author):

Please see attachment

Thank you for the opportunity for reviewing the revised manuscript by Seymour et al. Now entitled “Environmental DNA provides greater insight to biodiversity and ecosystem function

compared to traditional approaches, via spatio-temporal nestedness and turnover partitioning”. The motivation, main hypothesis, findings, and conclusions are clearer in the revised manuscript, particularly the revisions to the Introduction. I return to a point I made previously about ‘cost effectiveness’ below. I feel it an important consideration for this paper and needs, at least, clarification.

RESPONSE: Thank you for your time.

Alexander Fremier

I appreciate the time and thought that went into the re-write of the introduction. The revisions satisfy my multiple comments about the refocusing the manuscript on genetic based biodiversity techniques in monitoring, rather than a comparison paper. The transparent hypotheses R1 requested further support the revised manuscript focus. The figure adjustment also clarifies the data supporting discussion points. The first paragraph in the Discussion returns to the Main Points with clear and objective findings from the data. I think this paragraph will help the reader pull out the main findings. I think the revised manuscript has a clearer focus and structure.

RESPONSE: Thank you

I was probably not as clear in my initial review as I should have been. My comments on the cost efficiency/effectiveness is not about the amount of data produced by either method. Clearly, genetic based methods produce more data. The point here in the manuscript is about monetary cost. I have not read a paper that has compared genetic based versus field-based methods for cost efficiency. That is, for the same amount of data (e.g., benthic macroinvertebrate abundance data of indicator species only) which method is better? The authors argue, as I think I would as well, that genetic methods are more cost efficient than field methods. My point here is that a citation would be necessary to confirm this, or data from this work to show it. The later seems out of the scope of this paper. If there are no citations available, then perhaps the claim of cost efficiency should be clarified. This is obviously not a deal breaker for publication, but it is an important assumption that many pass over which has big implications for potential replacement monitoring techniques.

RESPONSE: We have removed the term “cost” from the abstract

Line 13 – “while reducing cost and time”. I want to believe this but I honestly do not think this direct claim is precise. Is there a cost comparison research paper out there yet? Not by my knowledge. Certainly, eDNA produces more data and data that field methods cannot produce. But from a strict monitoring needs driven by policy, field methods might produce the data the policy requires (e.g., high EPT taxa in a sample) at a low cost. Plus, eDNA has not been shown to reliably estimate population size. qPCR estimates are a start, but they cannot replace areal based estimates of field surveys. I do not see eDNA a replacement method yet, but it certainly has the potential. It is probably best described as a complementary method currently. The latter half of the Abstract is on point – it details the benefits and does not wade into the ‘cost’ complications that this paper does not directly address.

RESPONSE: The term “cost” has been removed from the abstract to alleviate concerns.

Line 140 – “better descriptor” perhaps change to “more complete”

RESPONSE: We have kept the term as more complete seems more vague

Line 268 – “With upstream transported limited above” - typo? Overall, this sentence is difficult to understand and a very important point.

RESPONSE: Edited in line with reviewer 1’s suggestion

Line 330 to the end – Difficult recommendations to follow.

RESPONSE: Understood

Line 335 head water to headwater

RESPONSE: Edited, thank you

Line 338 to the end – Clear conclusion linking back to the Introduction.

RESPONSE: OK